# Sparse genetically defined neurons refine the canonical role of periaqueductal gray columnar organization

Mimi Q La-Vu[1,2], Ekayana Sethi[2], Sandra Maesta-Pereira[2], Peter J Schuette[1,2], Brooke C Tobias[2], Fernando MCV Reis[2], Weisheng Wang[2], Anita Torossian[1,2], Amy Bishop[3], Saskia J Leonard[2], Lilly Lin[2], Catherine M Cahill[3,4,5], Avishek Adhikari[2]*

[1]Neuroscience Interdepartmental Program, University of California, Los Angeles, Los Angeles, United States; [2]Department of Psychology, University of California, Los Angeles, Los Angeles, United States; [3]Hatos Center for Neuropharmacology, University of California, Los Angeles, Los Angeles, United States; [4]Department of Psychiatry and Biobehavioral Sciences, Los Angeles, United States; [5]Semel Institute for Neuroscience and Human Behavior, Los Angeles, United States

*For correspondence:
avi@psych.ucla.edu

Competing interest: The authors declare that no competing interests exist.

**Abstract** During threat exposure, survival depends on defensive reactions. Prior works linked large glutamatergic populations in the midbrain periaqueductal gray (PAG) to defensive freezing and flight, and established that the overarching functional organization axis of the PAG is along anatomically-defined columns. Accordingly, broad activation of the dorsolateral column induces flight, while activation of the lateral or ventrolateral (l and vl) columns induces freezing. However, the PAG contains diverse cell types that vary in neurochemistry. How these cell types contribute to defense remains unknown, indicating that targeting sparse, genetically-defined populations may reveal how the PAG generates diverse behaviors. Though prior works showed that broad excitation of the lPAG or vlPAG causes freezing, we found in mice that activation of lateral and ventrolateral PAG (l/vlPAG) cholecystokinin-expressing (CCK) cells selectively caused flight to safer regions within an environment. Furthermore, inhibition of l/vlPAG-CCK cells reduced predator avoidance without altering other defensive behaviors like freezing. Lastly, l/vlPAG-CCK activity decreased when approaching threat and increased during movement to safer locations. These results suggest CCK cells drive threat avoidance states, which are epochs during which mice increase distance from threat and perform evasive escape. Conversely, l/vlPAG pan-neuronal activation promoted freezing, and these cells were activated near threat. Thus, CCK l/vlPAG cells have opposing function and neural activation motifs compared to the broader local ensemble defined solely by columnar boundaries. In addition to the anatomical columnar architecture of the PAG, the molecular identity of PAG cells may confer an additional axis of functional organization, revealing unexplored functional heterogeneity.

## Editor's evaluation

The article is a tour de force examination of the role of PAG CCK neurons in threat. It is exemplary in the use of a variety of high- and low-threat tasks as well as gain and loss of CCK function approaches, and reporting of distinct behaviors. The results reported will be of significant benefit for those studying the behavioral and neural mechanisms of learned and unlearned fear and threat, and decision-making in threatening situations.

## Introduction

The midbrain periaqueductal gray (PAG) has been implicated in numerous functions, including pain modulation, vocalization, breathing, heart rate, hunting, freezing, and flight (*Behbehani, 1995*; *Keay and Bandler, 2015*; *Motta et al., 2017*; *Silva and McNaughton, 2019*). For decades, a great deal of effort has been put toward understanding how columnar subdivisions of the PAG control or contribute to distinct defensive behaviors (*Bandler et al., 1985*; *Bandler and Carrive, 1988*; *Bandler and Shipley, 1994*; *Carrive, 1993*; *de Andrade Rufino et al., 2019*; *Gross and Canteras, 2012*; *Leman et al., 2003*; *Morgan and Clayton, 2005*; *Tomaz et al., 1988*; *Walker and Carrive, 2003*; *Zhang et al., 1990*). Prior work indicates that the ventrolateral (vl) PAG column is necessary for conditioned freezing (*Tovote et al., 2016*). Though less studied than the vlPAG, optogenetic and electrical excitation of the lateral (l) PAG column also produces freezing (*Assareh et al., 2016*; *Bittencourt et al., 2005*; *Bittencourt et al., 2004*; *Yu et al., 2021*). The dorsolateral (dl) PAG has a key role in controlling innate defensive behaviors. Indeed, dlPAG cells encode numerous defense behaviors, including freezing, escape, and risk assessment (*Del-Ben and Graeff, 2009*; *Deng et al., 2016*; *Reis et al., 2021a*; *Reis et al., 2021b*), and activation of glutamatergic vGlut2+ dlPAG cells induces escape (*Evans et al., 2018*; *Tovote et al., 2016*). More recent work employing methods with genetic specificity has focused on large PAG populations positive for broadly expressed markers such as *Vgat, Vglut2,* or *CaMKIIa*. For example, optogenetic activation of PAG neurons expressing *CaMKIIa*, which is ubiquitous in the region, elicited both freezing and flight (*Deng et al., 2016*). Though these findings provided important insights, they leave open the question of whether sparser PAG populations might control and encode more specific behavioral metrics.

Indeed, the PAG contains a diverse array of sparse cell types (*Keay and Bandler, 2015*; *Silva and McNaughton, 2019*; *Yin et al., 2014*). These cell types exhibit different neurochemical profiles and vary in anatomical location, often spanning more than a single column (*Silva and McNaughton, 2019*). For example, substance P-producing *Tac1+* cells and enkephalin-releasing *Penk+* cells are concentrated in dorsomedial and ventrolateral posterior PAG, while somatostatin-expressing cells can be found in dorsomedial and lateral columns (*Allen Institute for Brain Science, 2021*; *Silva and McNaughton, 2019*). It is possible that distinct cell types contribute to specific phenotypes controlled by the PAG. Accordingly, genetically identified populations have been more deeply studied in other regions such as the lateral hypothalamus (*Li et al., 2018*) or the central amygdala (*Fadok et al., 2017*), leading to unprecedented insights on their function. However, cell-type-specific dissections of sparse PAG populations remain scarce, and the functions of specific molecularly defined cell populations are largely uncharacterized.

One population of interest is composed of cholecystokinin-releasing PAG (PAG-CCK) cells (*Allen Institute for Brain Science, 2021*). Intriguingly, intra-PAG infusion of CCK in rats induces defensive behaviors and potentiates one-way escape behavior (*Netto and Guimarães, 2004*; *Zanoveli et al., 2004*). Additionally, CCK excites PAG neurons at both pre- and postsynaptic loci (*Liu et al., 1994*; *Yang et al., 2006*), suggesting that PAG-CCK cells may have widespread effects on local cell activity dynamics. However, despite these tantalizing results, to date PAG-CCK cells have not been directly studied and their function remains unknown.

Here, we specifically target, manipulate, and monitor the neural activity of PAG-CCK cells. We show that lateral and ventrolateral (l/vl) PAG-CCK cells are a small subset of glutamatergic cells, and that they selectively control flight to a safe location within an environment without affecting other defensive behaviors such as freezing or other l/vlPAG-mediated processes such as analgesia. Furthermore, though decades of prior work has consistently shown that PAG cells are activated by proximity to danger (*Aguiar and Guimarães, 2009*; *Canteras and Goto, 1999*; *Deng et al., 2016*; *Evans et al., 2018*; *Mobbs et al., 2010*; *Mobbs et al., 2007*; *Watson et al., 2016*), we find that l/vlPAG-CCK cells are more active far from threat. In contrast, pan-neuronal activation of cells in the same l/vlPAG region induced freezing, and these cells closely encoded threat proximity and escape initiation. Thus, we show that characterization of sparser, genetically defined PAG populations may reveal cells that have unique functional roles and that may even show opposing patterns of neural activation relative to the broader local ensemble. Deciphering how molecularly defined PAG populations complement and interact with the well-established anatomical columnar functional framework is a key step in understanding how this ancient structure controls a constellation of vital behaviors.

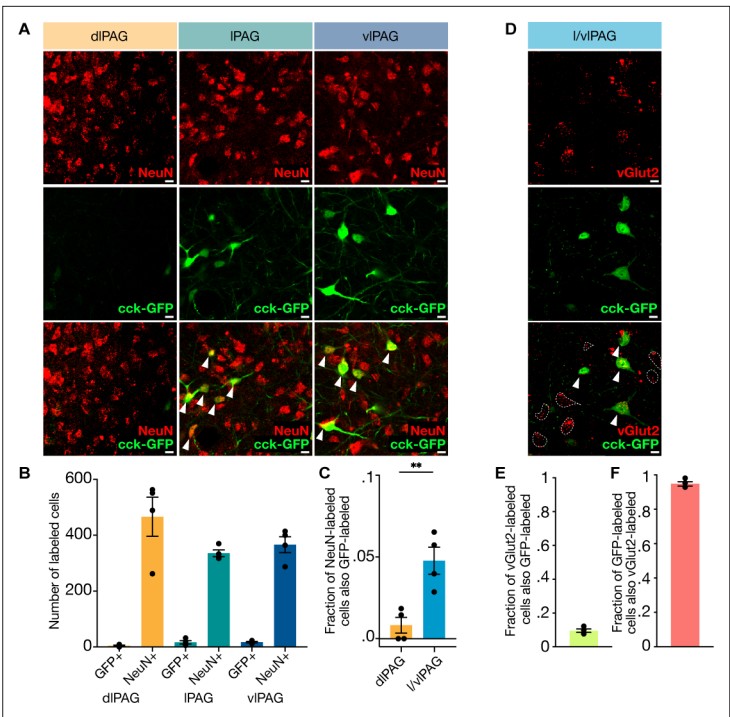

**Figure 1.** Cholecystokinin-expressing (CCK+) cells comprise approximately 5% of lateral/ventrolateral periaqueductal gray (l/vlPAG) neurons and are primarily glutamatergic. (**A**) Example histology images showing immunostaining of pan-neuronal marker NeuN (top row), viral-mediated expression of GFP in CCK-expressing cells (middle row), and overlay of NeuN and CCK-GFP (bottom row) in the dorsolateral (left column), lateral (middle column), and ventrolateral (right column) PAG. Scale bars, 10 µm. (**B**) Raw counts of CCK-GFP+ and NeuN+ cells in the dorsolateral PAG (dlPAG), lPAG, and vlPAG. (**C**) Fraction of NeuN-labeled cells that are also GFP-labeled in the dlPAG and l/vlPAG. CCK-expressing cells comprise ~5% of l/vlPAG neurons and constitute significantly more of l/vlPAG neurons than dlPAG neurons (n = 4; paired t-test, **P=.0032). (**D**) Immunostaining of glutamatergic marker vGlut2 in CCK cells. Example histology images showing vGlut2 (top), CCK-GFP (middle) and vGlut2/GFP overlay (bottom). White arrow indicates vGlut2+/GFP+ cell. Dashed outline indicates vGlut2+/GFP- cell. Scale bar, 10 µm. (**E**) 9.6% of vGlut2-labeled cells in the l/vlPAG are also GFP-labeled (n = 4; 302 vGlut2/GFP+ of 3115 vGlut2+ cells). (**F**) A majority (94.8%) of GFP-labeled cells in the l/vlPAG are also vGlut2-labeled (n = 4; 302 vGlut2+/GFP+ of 317 GFP+ cells). Errorbars: mean ± SEM.

The online version of this article includes the following figure supplement(s) for figure 1:

**Figure supplement 1.** In situ hybridization of vglut2 and cholecystokinin-expressing (CCK) cells in the lateral/ventrolateral periaqueductal gray (l/vlPAG) shows double labeling of vglut2 and CCK.

## Results
### CCK+ cells comprise a sparse glutamatergic subset of l/vlPAG neurons

The neuropeptide CCK is expressed primarily in two clusters within the PAG: one located in the dorso-medial column and one spanning both the lateral and ventrolateral columns (*Allen Institute for Brain Science, 2021*). Here, we focused on the latter population, which is more prevalent in the posterior than anterior PAG. To quantify the proportion of CCK+ neurons in the posterior l/vlPAG column, we used cre-dependent viral vectors to express GFP in CCK+ cells of *Cck-ires-cre* mice. We then immunostained posterior PAG slices against the pan-neuronal marker NeuN and quantified Neun/GFP overlap (*Figure 1A*). We observed GFP expression in the lPAG and vlPAG, but GFP expression was largely absent in the dlPAG (*Figure 1A, B*). Quantification showed that CCK-GFP cells comprise approximately 5% of l/vlPAG neurons and are more prevalent in the l/vlPAG than dlPAG (*Figure 1C*; n = 4, t (3) = 8.743, p=0.0032). CCK cells in several brain regions such as cortex, hippocampus, and amygdala are reported to be inhibitory (*Kepecs and Fishell, 2014*; *Mascagni and McDonald, 2003*; *Nguyen et al., 2020*; *Whissell et al., 2015*), though glutamatergic CCK+ cells have also been reported in other regions (*Wang et al., 2021a*). To determine if these cells are glutamatergic, we

immunostained against the glutamatergic marker vGlut2 in PAG slices containing GFP-expressing CCK+ cells (*Figure 1D*). We found that a small fraction (9.6%) of vGlut2-labeled cells were also GFP-labeled (*Figure 1E*). Similarly, in situ hybridizations revealed that 8.58% of vGlut2-expressing cells co-express CCK (*Figure 1*, *Figure 1—figure supplement 1*). Notably, we found that a majority of GFP-labeled cells (94.8%) were also vGlut2-labeled (*Figure 1F*). Our characterization shows that CCK+ cells comprise a small, sparse subset of PAG neurons that span the lateral and ventrolateral columns and are primarily glutamatergic.

## l/vlPAG-CCK stimulation induces a repertoire of behaviors distinct from pan-neuronal dlPAG and l/vlPAG stimulation

To study how various PAG populations may participate in distinct defensive phenotypes, we used an optogenetic approach to manipulate three different PAG subpopulations: pan-neuronal synapsin (syn)-expressing dorsolateral PAG neurons (dlPAG-syn), pan-neuronal syn-expressing lateral/ventro-lateral PAG neurons (l/vlPAG-syn), and cholecystokinin-expressing lateral/ventrolateral PAG neurons (l/vlPAG-CCK). We targeted these populations by local injection of adeno-associated viruses (AAVs) delivering channelrhodopsin-2 (ChR2) coupled to a yellow fluorescent protein (YFP) tag into the dlPAG or l/vlPAG of wildtype (WT) mice and l/vlPAG of *Cck-ires-cre mice* (*Figure 2A and B*, *Figure 2—figure supplement 1*). Mice injected with AAVs containing only YFP served as controls. The viral strategy used to transfect pan-neuronal l/vlPAG cells was synapsin-specific and did not exclude transfection of CCK+ cells. We first optogenetically manipulated naive mice in an open field (*Figure 2C–G*). Activation of dlPAG-syn cells increased speed and open-field corner entries compared to control mice (*Figure 2H and L*; dlPAG-eYFP, n = 5; dlPAG-ChR2, n = 4; speed: t(7) = 2.495, p=0.0413; corner entries: t(7) = 2.451, p=0.044). Notably, activation of only this, but not other PAG populations, induced escape jumping (*Figure 2I*; dlPAG-YFP, n = 5; dlPAG-ChR2, n = 4; t(7) = 6.111, p=0.0005). Light activation of l/vlPAG-syn cells strongly promoted freezing, and consequently reduced speed and corner entries (*Figure 2H, J and L*); (l/vlPAG-syn-eYFP, n = 5; l/vlPAG-syn-ChR2, n = 5; freezing: t(8) = 9.176, p<0.0001; speed: t(8) = 7.741, p < 0.0001; corner entries: t(8) = 4.548, p=0.0019). Finally, we observed that activation of l/vlPAG-CCK cells increased speed, reduced time spent in the open-field center, and increased corner entries (*Figure 2H, K, and L*; l/vlPAG-CCK-eYFP, n = 17; l/vlPAG-CCK-ChR2, n = 14; speed: t(29) = 3.667, p=0.001; center time: t(29) = 3.334, p=0.0023; corner entries: t(29) = 5.253, p<0.0001). Interestingly, activation of only this population increased time spent in the corners of the open field (*Figure 2M*; l/vlPAG-CCK-eYFP, n = 17; l/vlPAG-CCK-ChR2, n = 14; t(29) = 2.967, p=0.006). These results demonstrate that increased activity in these three PAG subpopulations elicited diverse behavioral phenotypes. Stimulation of l/vlPAG-CCK cells induced a repertoire of behaviors distinct from pan-neuronal l/vlPAG and dlPAG activation. Furthermore, l/vlPAG-CCK activation induced a preference for the corners of the open field, which represent the safest area in the arena as they allow mice to best limit visual detection by predators (*La-Vu et al., 2020*).

## l/vlPAG-CCK stimulation prompts entry into a dark burrow

We aimed to further investigate the exhibited preference for safety upon activation of l/vlPAG-CCK cells. We developed the latency to enter (LTE) assay, a novel paradigm that measures flight to the safest region within an environment. The LTE is a square arena illuminated to 80 lux and contains a dark burrow (2 lux) in one corner. Mice were habituated to the arena for 10 min. To verify that mice perceived the burrow as a safer area within the assay, only mice that exhibited a preference for the burrow over the other three corners during habituation continued to test on the following day (91.3%; 63 of 69 mice showed burrow preference). During test, mice were placed in the LTE for a 1 min context reminder prior to 10 consecutive trials. For optogenetic manipulation within the LTE, light delivery was alternated across the 10 trials, beginning with a light-off trial. Prior to the start of each trial, mice were confined to the corner opposite of the burrow, the holding zone, with a transparent barrier for 15 s. For light-on trials, light was delivered for the last 5 s of the 15 s period in the holding zone and continued until the end of the trial. The start of a trial (after 15 s in holding zone) was marked by barrier removal and the trial ended upon burrow entry or 60 s had passed. If a mouse entered the burrow, they could remain in the burrow for 10 s prior to being returned to the holding zone. If they did not enter, they were immediately returned to the holding zone. The LTE allows assessment of the LTE a burrow from a fixed start location within an arena and enables structured sampling across

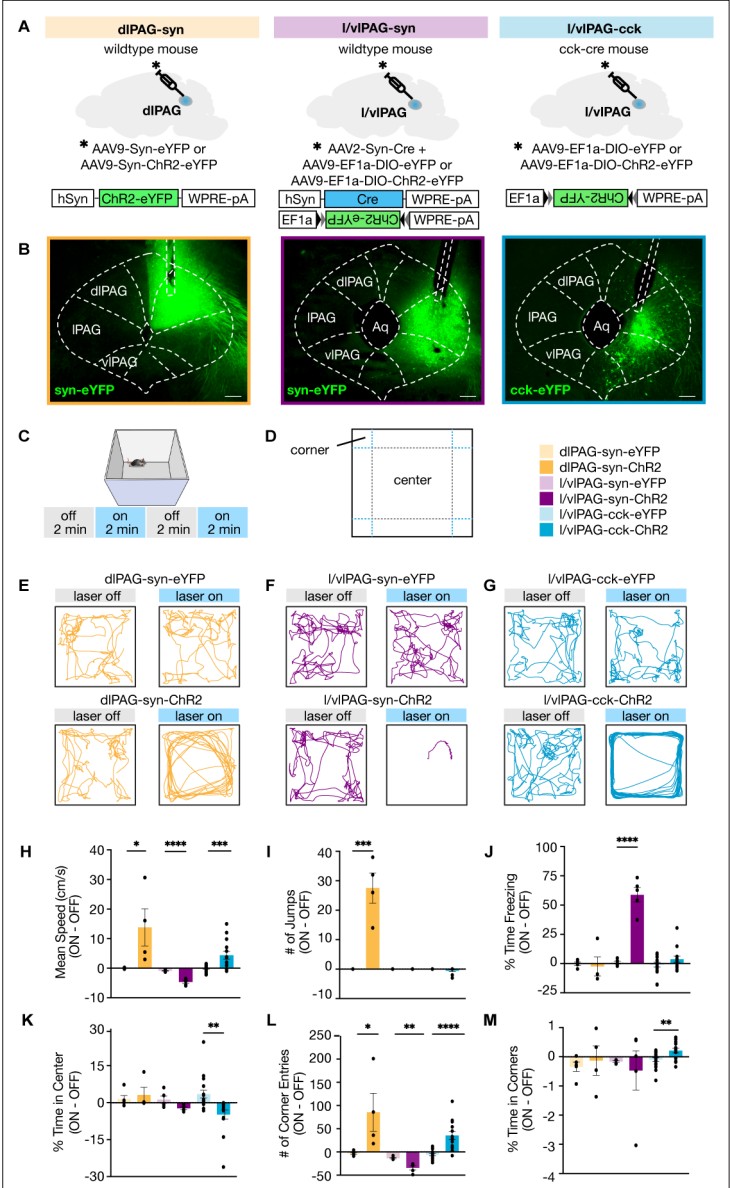

**Figure 2.** Lateral/ventrolateral periaqueductal gray-cholecystokinin-expressing (l/vlPAG-CCK) cell stimulation induced a repertoire of behaviors distinct from pan-neuronal l/vlPAG and dorsolateral PAG (dlPAG) activation. (**A**) Viral strategy to express enhanced yellow fluorescent protein (eYFP) or light-sensitive channelrhodopsin (ChR2-eYFP) in synapsin-expressing cells in the dlPAG (dlPAG-syn, left), synapsin-expressing cells in the l/vlPAG (l/vlPAG-syn, middle), and cholecystokinin-expressing cells in the l/vlPAG (l/vlPAG-CCK, right). A fiber optic cannula was then implanted over respective regions. (**B**) Histology of eYFP expression in dlPAG-syn (left), l/vlPAG-syn (middle), and l/vlPAG-CCK (right). Scale bar, 200 µm. (**C**) Stimulation protocol in the open field. Blue light (473 nm, 5 ms, 20 Hz) was delivered in alternating 2 min epochs (OFF-ON-OFF-ON) for 8 min total. (**D**) Diagram indicating center and corners of the open-field assay. (**E–G**) Example locomotion maps in the open field during laser-on and laser-off epochs of either mice expressing eYFP (top) or ChR2-eYFP (bottom) in dlPAG-syn (**E**), l/vlPAG-syn (**F**), and l/vlPAG-CCK (**G**) populations. (**H–M**) Bars depict respective behaviors during light-off epochs subtracted from light-on epochs (ON minus OFF). Light delivery to dlPAG of syn-ChR2 mice increased mean speed, jumps, and corner entries compared to dlPAG-syn-eYFP mice (dlPAG-eYFP, n = 5; dlPAG-ChR2, n = 4; unpaired t-tests; speed, *p=0.041; jumps, ***p=0.0005; corner entries, *p=0.044). Light delivery to the l/vlPAG of syn-ChR2 mice reduced mean speed, increased freezing, and reduced corner entries compared to l/vlPAG-syn-eYFP mice (l/vlPAG-syn-eYFP, n = 5; l/vlPAG-syn-ChR2, n = 5; unpaired t-tests; speed, ****p<0.0001; freezing, ****p<0.0001; corner entries, **p=0.0019). Light delivery to the l/vlPAG of CCK-ChR2 mice increased mean speed and corner entries while reducing center time compared to l/vlPAG-CCK-eYFP mice (l/vlPAG-CCK-eYFP, n = 17; l/vlPAG-CCK-ChR2, n =

*Figure 2 continued on next page*

*Figure 2 continued*

14; unpaired *t*-tests; speed, ***p=0.001; corner entries, ****p<0.0001; center time, **p=0.0023). Importantly, time spent in corners increased during light delivery to l/vlPAG-CCK-ChR2 mice compared to l/vlPAG-CCK-eYFP mice (unpaired *t*-test, **p=0.006). Errorbars: mean ± SEM.

The online version of this article includes the following figure supplement(s) for figure 2:

**Figure supplement 1.** ChR2 expression and fiber placement in the periaqueductal gray (PAG) in coronal brain sections.

regular trials that is not dependent on mice traversing to a start location to initiate a trial, minimizing variability in inter-trial intervals.

To study if activation of PAG subpopulations can bias mice to flee to the burrow, we optogenetically manipulated dlPAG, l/vlPAG, and l/vlPAG-CCK neurons in the LTE assay (*Figure 3A and B*). Despite similar levels of preference for the burrow during habituation, only optogenetic activation of l/vlPAG-CCK neurons reduced LTE the burrow relative to YFP control mice (*Figure 3C–F*; l/vlPAG-CCK-eYFP, n = 17; l/vlPAG-CCK-ChR2, n = 14; t(29) = 4.108, p=0.0003). Notably, activation of syn-expressing l/vlPAG neurons robustly increased latency as l/vlPAG-syn-ChR2 mice displayed substantial freezing with light-delivery compared to YFP mice (*Figure 3F*; l/vlPAG-syn-eYFP, n = 5; l/vlPAG-syn-ChR2, n = 5; t(8) = 3.777, p=0.0054). These data demonstrate that increased activity in l/vlPAG-CCK neurons can induce urgent flight to a safe burrow in a low-threat environment, a feature distinct from pan-neuronal l/vlPAG and dlPAG cells.

## l/vlPAG-CCK stimulation is aversive and anxiogenic, and can induce a hallmark sympathetic response

As there are no reports of genetically targeted manipulation of l/vlPAG-CCK cells, we sought to further characterize the behavioral phenotype induced by l/vlPAG-CCK activation. We assessed the effects of optogenetic activation in mice expressing ChR2 in l/vlPAG-CCK cells compared to YFP controls in anxiety and defense-related assays. Pairing light activation of l/vlPAG-CCK cells with one of two chambers in the real-time place test assay resulted in avoidance of the stimulated chamber, suggesting increased l/vlPAG-CCK activity is aversive (*Figure 4A–C*; eYFP, n = 14; ChR2, n = 8; t(20) = 4.938, p<0.0001). Furthermore, stimulation of l/vlPAG-CCK cells in the elevated plus maze (EPM) reduced time spent in the open arms of the maze (*Figure 4D–F*; eYFP, n = 16; ChR2, n = 10; t(24) = 3.391, p=0.0024). A majority of laser onsets of the fixed duration stimulation protocol occurred while ChR2 mice occupied a closed arm (60.0% ± 10.0%, n = 10); the same mice spent a majority of stimulation epochs in a closed arm (68.16% ± 3.39%), excluding the possibility that arm occupancy at laser onset may result in aversion of said arm. Light activation of l/vlPAG-CCK cells also markedly increased pupil size, a hallmark sympathetic response (*Figure 4G and H*; eYFP, n = 4; ChR2, n = 7; t(9) = 2.908, p=0.0174). Pupil size measurements were carried out at a lower laser intensity (1.5 mW versus 3.5 mW in behavioral experiments) to avoid overt motor changes during head fixation as well as movement-related arousal or stress that may have confounded measurements. Together, these results suggest that l/vlPAG-CCK cell activation is aversive, anxiogenic, and may elicit sympathetic activation.

## l/vlPAG-CCK inhibition delays entry into a dark burrow

Our data show that activation of l/vlPAG-CCK neurons is sufficient to drive flight to safety (*Figure 3*). To determine if these neurons serve a critical role in these conditions, we next used AAV-mediated, cre-dependent bilateral expression of the inhibitory opsin archaerhodopsin (Arch) in *Cck-ires-cre* mice to optically suppress activity of l/vlPAG-CCK cells in the LTE assay (*Figure 5A and B*, *Figure 5—figure supplement 1*). During test, green light (562 nm, constant) was delivered to the l/vlPAG in alternating trials and the LTE the burrow was measured at the end of each trial (*Figure 5B*). Though burrow preference was similar across both groups during habituation, light inhibition of l/vlPAG-CCK cells increased LTE the burrow in Arch mice compared to eGFP control mice (*Figure 5C–F*; GFP, n = 6; Arch, n = 7; t(11) = 2.447, p=0.0324). Thus, in addition to our activation studies, we show activity in l/vlPAG-CCK cells can bidirectionally control flight to a dark burrow under low-threat conditions. l/vlPAG-CCK stimulation increases avoidance of a live predator.

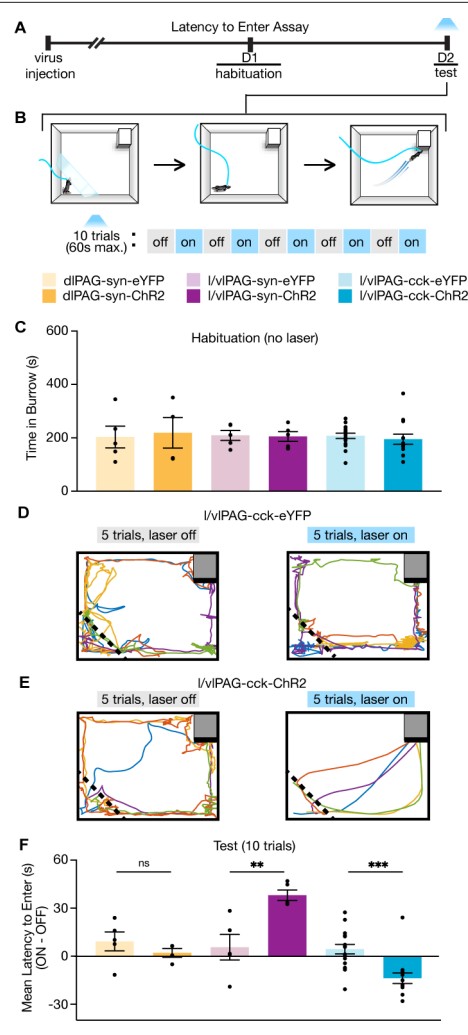

**Figure 3.** Lateral/ventrolateral periaqueductal gray-cholecystokinin-expressing (l/vlPAG-CCK) cell stimulation prompts entry into a dark burrow in the absence of threat, unlike pan-neuronal l/vlPAG and dorsolateral PAG (dlPAG) stimulation. (**A**) Timeline of latency to enter assay. Mice were habituated to a chamber containing a dark burrow for 10 min 1 day prior to test. Mice with preference for burrow during habituation were included during test. (**B**) Schematic of latency to enter assay during test. Left: at the beginning of each trial, mice were confined to a corner opposite of a dark burrow with a transparent wall (holding zone). After 15 s, the transparent barrier is removed and mice can freely move about the chamber. The trial ends when mice enter the burrow or 60 s have passed. Blue light (473 nm, 5 ms, 20 Hz) is delivered in alternating trials. In light-on trials, blue light delivery begins 5 s prior to barrier removal. After burrow entry, mice can remain in the burrow for 10 s before being returned to the holding zone. (**C**) Bars represent average time spent in the burrow during a 10 min habituation (dlPAG: eYFP, n = 5; ChR2, n = 4; l/vlPAG-syn: eYFP, n = 5; ChR2, n = 5; l/vlPAG-CCK: eYFP,

*Figure 3 continued*

n = 17, ChR2, n = 14). (**D, E**) Example locomotion map of five light-off (left) and five light-on trials (right) in a l/vlPAG-CCK-eYFP mouse (**D**) and l/vlPAG-CCK-ChR2 mouse (**E**). (**F**) Individual dots represent mean latency during five light-on epochs subtracted by mean latency during five light-off epochs (ON minus OFF). Trials without entry were regarded as latency of 61 s. Light delivery increased latency to enter the burrow in l/vlPAG-syn-ChR2 mice compared to l/vlPAG-syn-eYFP mice (l/vlPAG-syn-eYFP, n = 5; l/vlPAG-syn-ChR2, n = 5; unpaired *t*-test, **p=0.0054). Light delivery reduced latency to enter the burrow in l/vlPAG-CCK-ChR2 mice compared to l/vlPAG-CCK-YFP mice (l/vlPAG-CCK-eYFP, n = 17; l/vlPAG-CCK-ChR2, n = 14; unpaired *t*-test, ***p=0.0003). Stimulation of the dlPAG did not affect latency (dlPAG-eYFP, n = 5; dlPAG-ChR2, n = 4). Errorbars: mean ± SEM.

Our data show that l/vlPAG activity is sufficient and necessary for flight to safety in the LTE, a low-threat environment in which perceived danger is diffuse and uncertain (*La-Vu et al., 2020*). However, it is still unknown if the l/vlPAG-CCK population is involved in flight to safety in the presence of a well-defined, proximal threat such as a live predator.

To address this question, we optogenetically activated l/vlPAG-CCK cells while introducing mice to live predator exposure (*Figure 6A*). In this assay, mice are placed in an elongated rectangular arena that contains an awake rat restrained to one end by a harness (*Reis et al., 2021a*; *Wang et al., 2021a*; *Wang et al., 2021b*). Rats are natural predators of mice, and mice exhibit robust defensive reactions during exposure to a live rat but not a similarly shaped toy rat such as increased freezing, increased distance from the live rat, and reduced time in the zone containing the live rat (*Figure 6B*; n = 10; freezing: t(9) = 3.519, p=0.0065; threat distance: t(9) = 13.09, p<0.0001; threat zone: t(9) = 7.604, p<0.0001; *Wang et al., 2021b*). As the chamber does not contain a barrier and mice can freely roam the entire arena, live predator exposure elicits a naturalistic and diverse repertoire of defensive responses (*Reis et al., 2021a*; *Wang et al., 2021b*).

We hypothesized that activation of l/vlPAG-CCK cells in the presence of a live predator would exacerbate avoidance of the threat. To test this hypothesis, we delivered blue light (473 nm, 5 ms, 20 Hz) to the l/vlPAG of mice expressing YFP or ChR2-YFP in CCK+ cells during live predator exposure (*Figure 6C*). Light was delivered in alternating 2 min epochs (*Figure 6D and E*). Light activation of CCK+ cells reduced time spent in

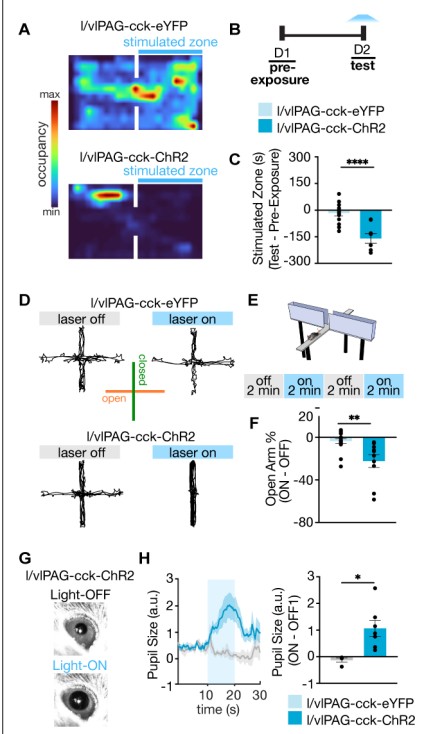

**Figure 4.** Further characterization of lateral/ventrolateral periaqueductal gray-cholecystokinin-expressing (l/vlPAG-CCK) neurons demonstrates stimulation is aversive, anxiogenic, and induces hallmark sympathetic responses. (**A**) Example spatial map of real-time place test (RTPT) depicting min/max occupancy during test of a l/vlPAG-CCK-eYFP mouse (top) and l/vlPAG-CCK-ChR2 mouse (bottom). Blue light delivery was paired with occupancy of one chamber in the RTPT during the 10 min test. (**B**) Timeline for RTPT assay. Each session lasted 10 min. (**C**) Dots represent time spent in the stimulated zone during test minus time spent in the same zone during pre-exposure (without light delivery). Bars are averaged across l/vlPAG-CCK-eYFP and l/vlPAG-CCK-ChR2 groups, respectively. Stimulation of l/vlPAG-CCK neurons results in avoidance of stimulated zone, compared to control YFP group (eYFP, n = 14; ChR2, n = 8; unpaired *t*-test, ****p<0.0001). (**D**) Example locomotion maps of l/vlPAG-CCK-eYFP (top) and l/vlPAG-CCK-ChR2 mice (bottom) during light-off (left) and light-on (right) epochs. (**E**) Stimulation protocol in elevated plus maze (EPM) assay. (**F**) Dots represent percent of time spent in open arms during light-on epochs normalized by light-off epochs (ON minus OFF) of l/vlPAG-CCK-eYFP or l/vlPAG-CCK-ChR2 mice. Light delivery to ChR2 mice reduced open-arm occupancy relative to eYFP mice (eYFP, n = 16; ChR2, n = 10; unpaired *t*-test, **p=0.0024). (**G**) Example pupil images of a head-fixed l/vlPAG-CCK-ChR2 mouse without (top) and with blue-light delivery (bottom). (**H**) Left: average data showing pupil size during baseline, stimulation, and post-stimulation periods (labeled OFF, ON, and

*Figure 4 continued*

OFF, respectively). Each period lasted 10 s. During stimulation, blue light was delivered to l/vlPAG. Right: blue light delivery increased pupil size in l/vlPAG-CCK-ChR2 compared to l/vlPAG-CCK-eYFP mice (eYFP, n = 4; ChR2, n = 7; unpaired *t*-test, *p=0.0174). Errorbars: mean ± SEM.

the threat zone and increased distance from the live rat (*Figure 6F and G*; eYFP, n = 10; ChR2, n = 9; time in threat zone: t(17) = 3.808, p=0.0014; distance: t(17) = 3.24, p=0.0048). Mice exhibit increased stretch-attend postures during exposure to predatory rats (*Wang et al., 2021b*). This measure was reduced as a result of optogenetic activation, demonstrating that not all defensive behaviors are promoted by CCK+ cell activation (*Figure 6I*; eYFP, n = 10; ChR2, n = 9; t(17) = 2.441, p=0.0259). Optogenetic activation of CCK+ cells also induced a trend toward reduced number of approaches toward the rat and did not alter freezing or locomotion (*Figure 6H, J, and K*; eYFP, n = 10; ChR2, n = 9; approaches: t(17) = 1.965, p=0.066; freezing: t(17) = 0.4696, p=0.6446; locomotion: t(17) = 1.682, p=0.1109). Escape velocity can be an informative measure of threat avoidance; however, ChR2 mice did not exhibit enough escapes during light activation to compute this measure as they did not consistently approach the rat (see representative exploration track in *Figure 6E*, bottom row), which decreased escapes from the rat as escapes cannot occur without prior approach. These results demonstrate that activation of l/vlPAG-CCK cells selectively enhanced avoidance of a live predator without altering freezing.

Importantly, these same results were not observed with CCK+ activation during exposure to a control toy rat (*Figure 6*, *Figure 6—figure supplement 1*). Activating CCK+ cells in this condition induced the same type of thigmotaxis seen during CCK+ activation in an open field (*Figure 2G*). In the presence of the toy rat, thigmotaxis was uniformly induced throughout the environment periphery, both near and far away from the toy rat (see representative exploration track in *Figure 6*, *Figure 6—figure supplement 1A*). Thus, in the presence of the toy rat, CCK+ activation induced avoidance of open spaces, rather than avoidance of the toy rat. In contrast, in the presence of the rat, activation of CCK+ cells induced thigmotaxis only in the corners furthest away from the live rat (*Figure 6E*). These data show that l/vlPAG-CCK cell activation increases avoidance of a live predator, but not of a control

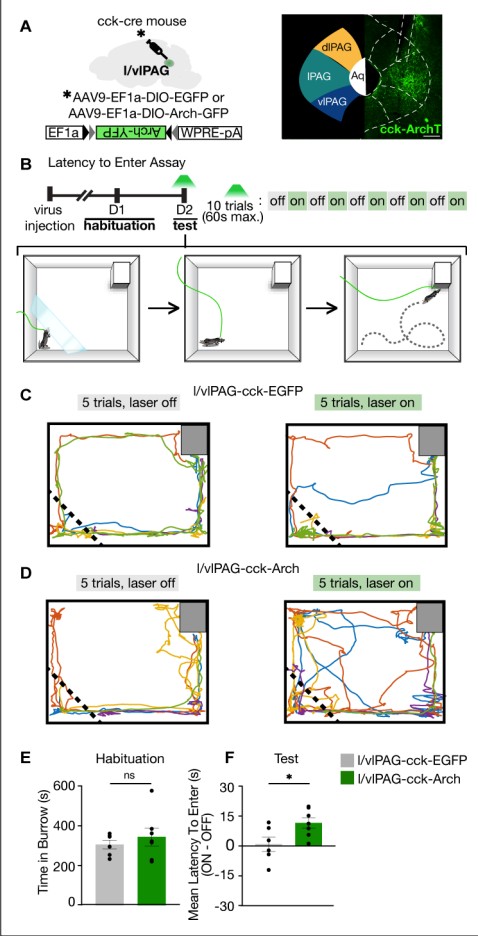

**Figure 5.** Lateral/ventrolateral periaqueductal gray-cholecystokinin-expressing (l/vlPAG-CCK) inhibition delays entry into a dark burrow. (**A**) Left: strategy for viral expression of cre-dependent GFP or Arch-GFP in l/vlPAG of CCK-cre mice. Right: histology showing Arch-GFP expressed in CCK+ cells in the l/vlPAG. Scale bar, 200 µm. (**B**) Top: timeline of latency to enter assay. Test consists of 10 trials, with green light delivery to l/vlPAG in alternating trials. Bottom: schematic of assay during test. At the start of trial, mice were confined to a holding zone for 15 s with a transparent barrier. When the barrier was removed, mice were free to explore the arena. Trial ended upon burrow entry or 60 s have passed. (**C**) Example locomotion maps of five trials without (left) and with (right) green light delivery in l/vlPAG-CCK-GFP mice. (**D**) Same as (**C**) but in l/vlPAG-CCK-Arch mouse. (**E**) No difference in burrow occupancy during 10 min habituation between l/vlPAG-CCK-GFP and l/vlPAG-CCK-Arch mice (GFP, n = 6; Arch, n = 7; unpaired *t*-test). (**F**) Green light delivery to l/vlPAG increased latency to enter burrow in l/vlPAG-CCK-Arch mice compared to l/vlPAG-CCK-GFP mice. Each dot represents average latency during five light-on trials minus average latency of five light-off trials (GFP, n = 6; Arch, n = 7; unpaired *t*-test, *p=0.0324). Errorbars mean ± SEM.

*Figure 5 continued on next page*

---

*Figure 5 continued*

The online version of this article includes the following figure supplement(s) for figure 5:

**Figure supplement 1.** Bilateral fiber placement for optogenetic inhibition in coronal brain sections.

---

safe toy rat, showing that these cells may serve to minimize threat exposure by directing exploration toward safer regions within an environment.

## l/vlPAG-CCK inhibition reduces avoidance of a live predator

To evaluate the necessity of l/vlPAG-CCK cells for threat avoidance in a high-threat environment, we bilaterally expressed cre-dependent inhibitory hM4di-mCherry in the l/vlPAG of *Cck-ires-cre* mice to chemogenetically inhibit CCK+ cells during live predator exposure (*Figure 7A*, *Figure 7—figure supplement 1A*). Mice expressing mCherry alone served as controls. A chemogenetic approach in this setting was beneficial because it enabled neuronal inhibition across a 10 min exposure without constant laser delivery as prolonged laser stimulation may induce tissue heating, among other issues (*Stujenske et al., 2015*). Both hM4di and mCherry-only mice were injected with clozapine-N-oxide (CNO) or saline prior to two exposures to a toy rat and two exposures to an awake, live rat on separate, sequential days (*Figure 7B and C*). Injections occurred 40 min prior to exposure and the order of drug delivery was counterbalanced across groups. All metrics were calculated as behavior exhibited following saline administration subtracted from behavior exhibited following CNO administration (CNO – SAL).

We found that l/vlPAG-CCK inhibition significantly increased time spent in the threat zone, increased the number of approaches toward the rat, and reduced escape velocity from the rat (*Figure 7D, F and H*; threat zone: mCherry, n = 8; hM4Di, n = 12; t(18) = 2.554, p=0.0199; approaches: mCherry, n = 8; hM4Di, n = 12; t(18) = 2.194, p=0.0496; escape velocity: mCherry, n = 7; hM4Di, n = 11; t(16) = 2.197, p=0.0431). CCK+ inhibition also induced a trend towards decreased distance from the rat (*Figure 7E*; mCherry, n = 8; hM4Di, n = 12; t(18) = 1.937, p=0.0686). Inhibition did not alter approach velocity, stretch-attend postures, freezing, or distance traveled (*Figure 7G,I–K*).

Importantly, l/vlPAG-CCK inhibition did not affect avoidance from a toy rat (*Figure 7*, *Figure 7—figure supplement 1B–I*), indicating

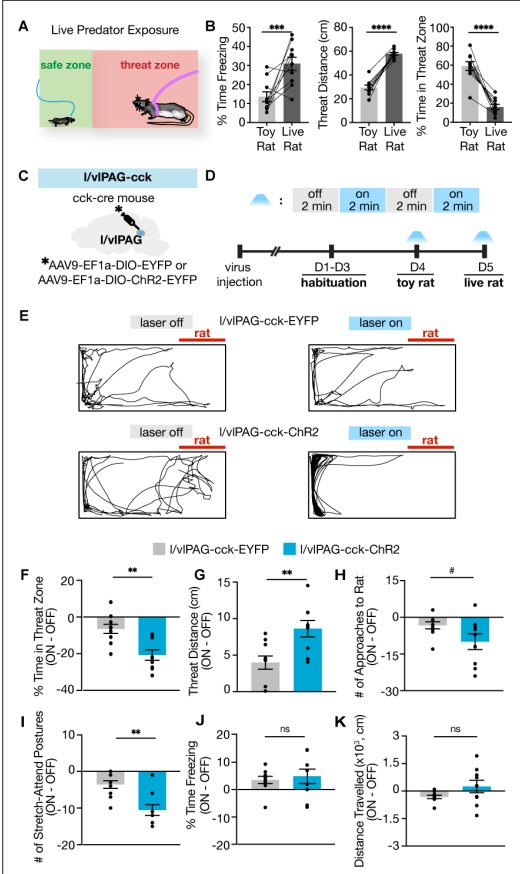

**Figure 6.** Lateral/ventrolateral periaqueductal gray-cholecystokinin-expressing (l/vlPAG-CCK) activation robustly enhances avoidance from a live predator without altering freezing. (**A**) Schematic of live predator exposure assay. Mice are placed in a long rectangular chamber (70 × 25 × 30 cm) containing an awake rat restrained with a harness to one end. The chamber does not contain a barrier and mice can move freely. The area containing the rat is considered a 'threat zone,' and the area furthest from the rat is considered a 'safe zone.' (**B**) Exposure to a live rat increased freezing and threat distance while reducing time in threat zone compared to exposure to a toy rat (n = 10, paired t-tests; freezing, ***p=0.0065; threat distance, ****p<0.0001; time in threat zone, ****p<0.0001). (**C**) EYFP or ChR2-eYFP was expressed in l/vlPAG-CCK cells and a fiber-optic cannula was implanted over the l/vlPAG. (**D**) Timeline of live predator assay. Blue light was delivered in alternating 2 min epochs during toy rat exposure and live rat exposure. (**E**) Example locomotion maps during laser-off (left) and laser-on (right) epochs of an eYFP mouse (top) and ChR2-eYFP mouse (bottom). (**F–K**) Optogenetic stimulation of l/vlPAG-CCK cells reduced time in threat zone (**F**; eYFP, n = 10; ChR2, n = 9; unpaired t-test, **p=0.0014), increased threat distance (**G**, unpaired t-test, **p=0.0048), and reduced stretch-attend postures (**I**, unpaired t-test, **p=0.0012). Number of approaches to the rat trended toward significance (**H**, #p=0.066). Freezing (**J**) and

*Figure 6 continued on next page*

*Figure 6 continued*

distance traveled (**K**) were not significantly affected. Errorbars: mean ± SEM.

The online version of this article includes the following figure supplement(s) for figure 6:

**Figure supplement 1.** Optogenetic activation of lateral/ventrolateral periaqueductal gray-cholecystokinin-expressing (l/vlPAG-CCK) neurons during toy rat exposure.

that the effects of inhibition are specific to a live predator. Inhibition of CCK cells also did not affect pain response latency during exposure to a heated plate assay (*Figure 7—figure supplement 2*), demonstrating that these cells do not affect other PAG functions such as analgesia (*Samineni et al., 2017*). Our data also show that inhibition of CCK+ cells did not alter learning of auditory cued conditioned fear in the experimental conditions used (*Figure 7—figure supplement 2*). However, this negative result may be due to ceiling levels of freezing, and it is possible that these cells may have a role in fear learning employing different protocols. Together, these results show that, in addition to controlling avoidance measures under low-threat conditions, l/vlPAG-CCK cells also selectively and bidirectionally control avoidance measures from a high-threat predator.

## l/vlPAG-syn cells are more active near threat, while l/vlPAG-CCK cells are more active far from threat

Numerous prior reports have consistently shown that PAG cells are activated by proximity to danger (*Deng et al., 2016*; *Evans et al., 2018*; *Mobbs et al., 2010*; *Mobbs et al., 2007*; *Reis et al., 2021a*; *Watson et al., 2016*). We next sought to observe endogenous l/vlPAG-CCK activity under both low- and high-threat conditions. We performed in vivo fiber photometry recordings of synapsin and CCK-expressing neurons in the l/vlPAG in the EPM and live predator exposure assay (*Figure 8A–C*, *Figure 8—figure supplement 1*). These assays offer a safety gradient that allows us to assess how population activity is spatially modulated by threat proximity. Recording of syn-expressing l/vlPAG cells will inform whether activity patterns in CCK-only population recordings are cell type-specific or region-specific.

We found that syn-GCaMP6f and CCK-GCaMP6f activity was differentially modulated by EPM arms (*Figure 8D*). Specifically, mean df/F of

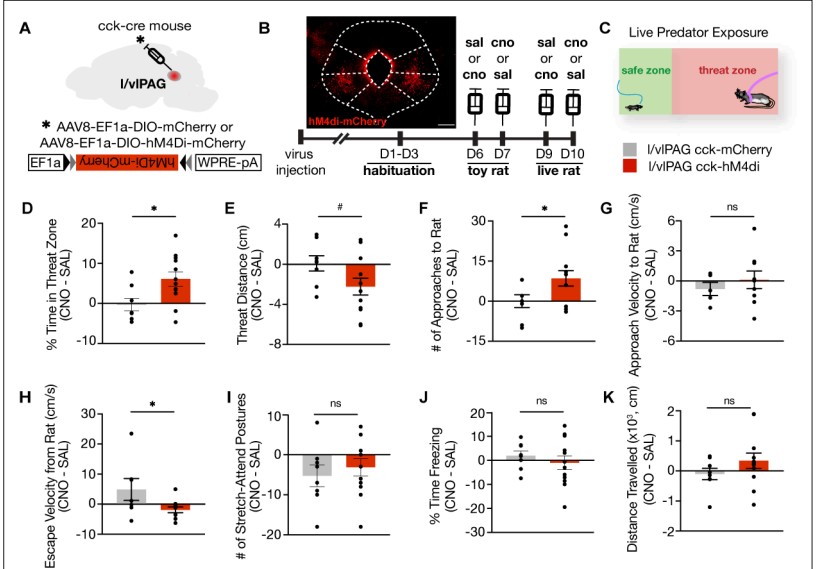

**Figure 7.** Lateral/ventrolateral periaqueductal gray-cholecystokinin-expressing (l/vlPAG-CCK) inhibition increases time spent near a live predator and reduces escape vigor without altering freezing. (**A**) Viral strategy for bilateral expression of inhibitory designer receptor hM4Di-mCherry or mCherry in CCK cells in the l/vlPAG. (**B**) Top-left: expression of hM4Di-mCherry in l/vlPAG-CCK cells. Scale bar, 200 μm. Bottom: timeline for DREADD experiments. Saline or clozapine-N-oxide (CNO, 10 mg/kg) occurred 40 min prior to exposure. (**C**) Live predator exposure schematic. Mice were placed in the presence of an awake rat restrained to one end of a chamber. Each exposure lasted 10 min. (**D–K**) Chemogenetic inhibition of l/vlPAG CCK cells increased time spent in threat zone (**D**, unpaired *t*-test, *p=0.0199), increased number of approaches toward the rat (**F**, unpaired *t*-test, *p=0.0496), and reduced escape velocity from the rat (**H**, unpaired *t*-test, *p=0.0431). Threat distance trended toward significance with CCK inhibition (**E**, unpaired *t*-test, #p=0.069). Approach velocity (**G**), stretch-attend postures (**I**), freezing (**J**), and distance traveled (**K**) were unaltered with inhibition (**D–F, I–K**: mCherry, n = 8; hM4Di, n = 12; **G**: mCherry, n = 5, hM4Di, n = 9; **H**: mCherry, n = 7, hM4Di, n = 11). Errorbars: mean ± SEM.

The online version of this article includes the following figure supplement(s) for figure 7:

**Figure supplement 1.** Chemogenetic of lateral/ventrolateral periaqueductal gray-cholecystokinin-expressing (l/vlPAG-CCK) population using DREADDs during exposure to a toy rat.

**Figure supplement 2.** Inhibition of lateral/ventrolateral periaqueductal gray-cholecystokinin-expressing (l/vlPAG-CCK) neurons does not alter pain response latency or acquisition of learned fear.

l/vlPAG-syn cells increased after open-arm entry compared to closed-arm entry (*Figure 8E*; n = 9; t(8) = 2.856, p=0.0213). In contrast, CCK activity was greater following closed-arm entry compared to open-arm entry (*Figure 8F*; n = 11; t(10) = 2.561, p=0.0283). Thus, in the low-threat EPM, l/vlPAG-CCK activity diverged from broader l/vlPAG activity.

To determine if this feature might extend to a high-threat situation, we next performed photometry recordings of l/vlPAG pan-neuronal and CCK-only populations during live rat exposure (*Figure 8G*). Mean df/F within spatial bins of varying distances from the safe wall shows that syn-GCaMP6f activity was not differentially modulated when approaching the rat but significantly altered during escapes, with dF/F peaked when mice were most proximal to the predator and sharply reduced as mice gained distance from the predator (*Figure 8H*; approach: r(8) = –0.255, p=0.476; escape: r(8) = 0.932, p<0.0001). This pattern is consistent with previous reports coupling increased PAG activity with threat proximity and escape initiation (*Deng et al., 2016*; *Evans et al., 2018*; *Reis et al., 2021b*; *Watson et al., 2016*).

Conversely, in CCK-GCaMP6f mice, population activity was modulated during both approaches and escapes. CCK+ activity ramped down as mice moved closer to the rat and ramped up as mice escaped away from the rat (*Figure 8I*; approach: r(8) = –0.792, p=0.006; escape: r(8) = –0.703, p=0.023).

Syn-GCaMP6f activity was tightly time-locked with escapes, increasing prior to and peaking soon after escape initiation. CCK+ activity also increased prior to escape but exhibited sustained heightened activity post-escape onset (*Figure 8J and K*). Our optogenetic experiments showed that increasing

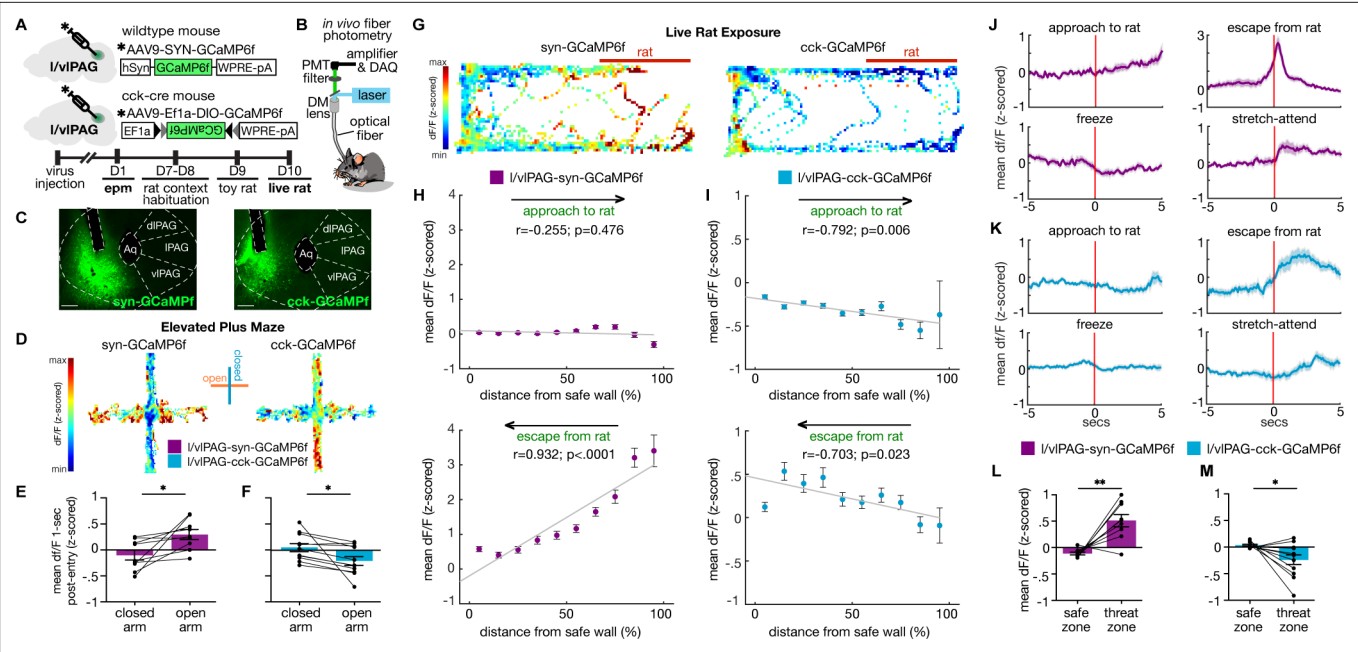

**Figure 8.** Lateral/ventrolateral periaqueductal gray-synapsin-expressing (l/vlPAG-syn) cells are more active near threat, while l/vlPAG-cholecystokinin-expressing (l/vlPAG-CCK) cells are more active far from threat. (**A**) Top: viral schematic for synapsin-specific and CCK-specific GCaMP6 expression in l/vlPAG. Bottom: timeline for in vivo photometry recordings. (**B**) Fiber photometry recording set-up. (**C**) Histology of GCaMP6f expression in synapsin-specific (left) and CCK-specific (right) cells in the l/vlPAG. Scale bar, 200 μm. (**D**) Example heatmaps showing z-scored dF/F in mice expressing synapsin-specific GCaMP6 (left) or CCK-specific GCaMP6 (right) in l/vlPAG in an elevated plus maze assay. Vertical arms of heatmaps represent closed arms. (**E, F**) Mean dF/F (z-scored) 1 s after arm entry in syn-GCaMP6 (**E**) and CCK-GCaMP6 mice (**F**). Mean dF/F 1 s post-entry into the open arms is greater than into closed arms in syn-GCaMP6 mice (**E**, n = 9; paired t-test, *p=0.0213), whereas mean dF/F 1 s post-entry into the open arms is lower than into the closed arms for CCK-GCaMP6 mice (**F**, n = 11; paired t-test, *p=0.0283). (**G**) Example heatmaps showing z-scored dF/F in syn-GCaMP6 (left) or CCK-GCaMP6 (right) in live rat exposure assay. Rat was confined to the right of the map, as indicated by the red bar. (**H, I**) Mean dF/F (z-scored) during approaches toward the rat (top) or escapes from the rat (bottom) within 10 spatial bins of varying distance from the safe wall of syn-GCaMP6 (**H**, n = 9) or CCK-GCaMP6 (**I**, n = 13) mice (syn-approach, n = 6744 samples; syn-escape, n = 2150 samples; CCK-approach, n = 7170 samples; CCK-escape, n = 2088 samples). (**H**) In syn-GCaMP6f mice, dF/F is positively correlated with distance from the safe wall during escapes from the predator (Pearson's correlation coefficient r = 0.932, p<0.0001). (**I**) In CCK-GCaMP6f mice, dF/F is negatively correlated with distance from safe wall during both approaches and escapes (approach, r = −0.792, p=0.006; escape, r = −0.703, p=0.023). (**J, K**) Mean dF/F (z-scored) 5 s before and after approaches, escapes, freeze bouts, and stretch-attend postures in syn-GCaMP6 (**L**) and CCK-GCaMP6 (**M**) populations (syn, n = 9; CCK, n = 13 for freeze, n = 12 for other behaviors). (**L, M**) Mean dF/F (z-scored) in the safer zone (one-third of assay near safer wall) and threat zone (two-thirds of assay distal from safer wall) in syn-GCaMP6 and CCK-GCaMP6 mice. Pan-neuronal l/vlPAG activity was increased in the threat zone compared to the safer zone (syn, n = 9; paired t-test, **p=0.0024), whereas CCK-specific activity was decreased in the threat zone compared to the safer zone (CCK, n = 12; paired t-test, *p=0.0223). Errorbars: mean ± SEM.

The online version of this article includes the following figure supplement(s) for figure 8:

**Figure supplement 1.** GCaMP6f expression and fiber placement in the lateral/ventrolateral periaqueductal gray (l/vlPAG) in coronal brain sections.

**Figure supplement 2.** No correlation between speed and df/F.

**Figure supplement 3.** Lateral/ventrolateral periaqueductal gray-synapsin-expressing (l/vlPAG-syn) and cholecystokinin (CCK) activity during exposure to a control toy rat.

activity in l/vlPAG-syn cells robustly induced freezing; however, we did not observe increased syn-GCaMP6f activity related to freeze bouts during predator exposure (**Figure 8J**). Finally, syn-GCaMP6f activity was greater in the threat zone than safe zone (**Figure 8L**; n = 9; t(8) = 4.375, p=0.0024) while CCK-GCaMP6f activity was decreased in the threat zone relative to the safe zone (**Figure 8M**; n = 12; t(11) = 2.658, p=0.0223).

Importantly, these effects were not due to a correlation between speed and df/F (**Figure 8—figure supplement 2**) and were not observed during exposure to a toy rat (**Figure 8—figure supplement 3**). Together, these results suggest that though the syn-expressing l/vlPAG population was particularly attuned to more threatening aspects of both the EPM and live predator exposure, CCK population

activity was negatively modulated by threat proximity and increased during escape movements away from threat.

## Discussion

Our study identifies a small, sparse, genetically defined subset of cells within the PAG, cholecystokinin-expressing neurons (CCK+) that control avoidance of threat in multiple contexts.

Activation of l/vlPAG CCK+ cells induced evasion of open spaces and a live predator, which are, respectively, low- and high-intensity threats. Conversely, inhibition of CCK+ cells delayed entry into a dark burrow and increased time spent near a live predator, showing that these cells bidirectionally control avoidance measures in both low- and high-threat environments. Importantly, our manipulations did not alter freezing in any condition, demonstrating a specific role for CCK+ cells in escape and avoidance. CCK+ cells also display increased activity with greater threat avoidance in both the low-threat EPM and the high-threat predator exposure.

We show that these features are specific to CCK+ cells and oppose the broader local ensemble, as pan-neuronal activation of the same lPAG and vlPAG region drove robust freezing (*Figure 2J*), and pan-neuronal activation patterns show increased activity with greater threat proximity (*Figure 8L*), consistent with prior reports (*Assareh et al., 2016*; *Bittencourt et al., 2005*; *Deng et al., 2016*; *Evans et al., 2018*; *Mobbs et al., 2010*; *Mobbs et al., 2007*; *Reis et al., 2021a*; *Tovote et al., 2016*; *Watson et al., 2016*; *Yu et al., 2021*). Importantly, though our strategy of synapsin-specific transfection does not exclude CCK+ cells, we show that CCK+ cells are a small, sparse population, making up only about 5% of l/vlPAG neurons (*Figure 1*), and are unlikely to be significantly driving fluorescence in pan-neuronal photometry recordings. Together, these data suggest that l/vlPAG CCK+ cells are selectively driving and signaling a behavioral state of threat avoidance and diverges from pan-neuronal l/vlPAG function.

We also characterized the effect of l/vlPAG CCK+ activation in pupil size, a physiological measure that is modulated by threat exposure (*Wang et al., 2021b*). To do so, we used lower blue laser light intensity (1.5 mW for pupil measurements versus 3–3.5 mW in all behavioral experiments). This was in order to avoid escape attempts, which may result in damage and dislocation of the fiber optic cannula when mice try to escape while head-fixed. These data suggest that lower l/vlPAG CCK+ stimulation is sufficient to produce a defensive state with physiological changes, while higher stimulation yields a more intense state that includes escapes, which are likely also accompanied by physiological changes.

One inconsistency in our findings is a lack of increase in pan-neuronal activity during freeze bouts in the predator assay (*Figure 8*), despite ChR2 activation of this population driving robust freezing in an open field (*Figure 2*) and LTE assay (*Figure 3*). This may be due to several factors including ChR2 activation may stimulate more ventral cells than are being recorded using fiber photometry. Another possibility is that ChR2 activation may drive activity in some cells that are responsible for driving the robust freezing observed, and the fluorescence of these sparse cells was washed out in Ca$^{2+}$ recordings. Finally, it is possible that l/vlPAG activity reflected complex population dynamics related to the heightened behavioral state induced by high-threat predator exposure, resulting in endogenous activity that is more complex than is elicited with artificial activation in a low-threat environment.

### l/vlPAG CCK cell activity may drive the threat-avoidance behavioral state

It is noteworthy that in all assays used in this work mice voluntarily chose when to avoid or approach threats, which consisted of either open spaces or a predator. Threat exposure is thus driven by internal state switches of threat approach and threat-avoidance states (*La-Vu et al., 2020*; *Reis et al., 2021b*). In the threat approach state, mice decrease distance to threats and perform risk-evaluation behaviors, such as stretch-attend postures. Conversely, in the threat-avoidance state, mice stay away from threats and initiate evasive escape.

We show that l/vlPAG CCK+ cells were active away from threats and during evasion from threats. Furthermore, activity of these cells was sufficient and necessary for threat avoidance and escape. We thus propose that l/vlPAG CCK+ cells are a key driver of the threat avoidance behavioral state. One intriguing question arising from this view is that if CCK+ activity causes escape when the mouse is near threat, then why does increased CCK+ activity in the safe region not elicit escape when the mouse is

away from threat as well? In other words, why did CCK+ activity not render the safe region aversive, and thus cause escape from this safe location? When the mouse approached the rat, it was in the threat approach state, and thus CCK+ activity was low. Immediately prior to escaping, the mouse switched to the threat-avoidance state and then escaped to the region far from the rat, and this action was accompanied by high CCK+ activity that presumably drove the escape. However, CCK+ activity remained high far from the rat in the safe zone. The 'safe region' in the rat assay was not truly danger-free, but was only safer in relation to the rat 'threat zone' (see scheme in *Figure 6A*). Even in the safe zone, it is likely that the mouse was motivated to further increase distance from the rat, as it was in the avoidance state, and this motivation may be related to the increased CCK+ activity seen away from the rat. However, there was no better or safer place for the mouse to occupy in this assay, so it remained in the 'safe zone' even though this region was still more aversive and dangerous than would be ideal for the mouse. This was also seen in our ChR2 activation of CCK+ cells during rat exposure, in which increased CCK+ activity did not cause indiscriminate escape from the safe zone, but rather strong thigmotaxis between the two safest corners within the safe zone. Thus, CCK+ activity did not cause escape from the safe zone because CCK+ activity is not predictive of simply escaping away from the current location, but rather it may increase the motivation to escape to the safest region within the environment. Furthermore, chemogenetic vl/lPAG CCK inhibition increased time near the predator, further supporting the view that CCK activity drives the threat avoidance state. Previously, we reported distinct synapsin-expressing PAG ensembles that consistently encode both threat approach and threat-avoidance states across different threat modalities, such as open spaces and predators (*Reis et al., 2021b*). This work suggests that CCK+ cells may be a genetically identified ensemble that promotes the threat-avoidance state during exposure to these same stimuli.

## Role of CCK cells in the l/vlPAG

Long-standing evidence links increased PAG activity with higher threat exposure. Following predator exposure, the rodent PAG exhibits increased Fos expression (*Aguiar and Guimarães, 2009*; *Canteras and Goto, 1999*; *Mendes-Gomes et al., 2020*). Pharmacological blockade of NMDA receptors in the dorsal or ventrolateral PAG increased open arm exploration in the EPM (*Guimarães et al., 1991*; *Molchanov and Guimarães, 2002*). Single-unit recordings show that dPAG and vPAG units display significant increases in firing rate after exposure to cat odor (*Watson et al., 2016*). Moreover, the dPAG is sensitive to sensory aspects of threat distance and intensity, displaying increased activity with greater proximity to an awake predator (*Deng et al., 2016*). In humans, PAG activity is positively correlated with threat imminence (*Mobbs et al., 2010*; *Mobbs et al., 2007*). Within the dPAG, glutamatergic neurons are key for escape initiation and vigor, and dPAG flight-related cells exhibited prominent firing early during flight and declined as mice fled further from a predator (*Deng et al., 2016*; *Evans et al., 2018*).

Similarly to previous reports of the PAG (*Assareh et al., 2016*; *Bittencourt et al., 2005*; *Tovote et al., 2016*; *Yu et al., 2021*), we also observed higher pan-neuronal PAG activity during threat proximity. However, in contrast, CCK+ activity was reduced with threat proximity. To our knowledge, this is the first PAG cell type identified in which endogenous activity is reduced with threat proximity.

Pan-neuronal optogenetic activation of l/vlPAG cells produced strong freezing (*Figure 2J*), in agreement with prior reports showing that electrical or optogenetic excitation of the lPAG or vlPAG produced freezing (*Assareh et al., 2016*; *Bittencourt et al., 2005*; *Bittencourt et al., 2004*; *Yu et al., 2021*). Furthermore, activation of glutamatergic vlPAG neurons powerfully elicited freezing (*Tovote et al., 2016*), and vlPAG lesions curtailed freezing during predator exposure and conditioned fear (*de Andrade Rufino et al., 2019*; *Fanselow et al., 1995*). Taken together, a plethora of data link lPAG and vlPAG activation with freezing. The identification of l/vlPAG CCK+ neurons as a population that drives escape, rather than freezing, thus opposes the canonical function of this region.

Prior data have suggested that the PAG participates in relaying aversive unconditioned stimulus information to the amygdala to inform associative plasticity, and this feature is critical to prediction error coding (*Herry and Johansen, 2014*; *Johansen et al., 2010*; *McNally et al., 2011*). Though studies have identified a role of vlPAG neurons in prediction error coding (*Ozawa et al., 2017*; *Walker et al., 2020*), our inhibition of CCK+ cells during fear acquisition did not alter freezing during acquisition nor retrieval, suggesting that a role in prediction error coding may be carried out by other vlPAG cells. It is, however, possible that these cells have a role in controlling learned fear under other

conditions that were not tested in this work. Our l/vlPAG activation studies reaffirm the lPAG and vlPAG roles in freezing and concurrently highlight the non-canonical role of CCK+ cells in driving flight.

The PAG's role in flight has historically been attributed to the dorsal PAG. Dorsal PAG stimulation in rodents has been shown to induce marked escape responses such as explosive vertical jumping, running, aversion, and panic-related sympathetic responses (*Del-Ben and Graeff, 2009*; *Depaulis et al., 1992*; *Fanselow, 1991*; *Jenck et al., 1995*; *Perusini and Fanselow, 2015*; *Valenstein, 1965*). The present findings suggest the role of the PAG in avoidance extends beyond the dorsal PAG column, in part, to the l/vlPAG. Interestingly, the flight pattern observed from l/vlPAG CCK+ activation is characteristically different from escape canonically described in PAG studies as they are void of robust protean vertical jumping seen in dlPAG activation in this study (*Figure 2I*) and prior work (*Ullah et al., 2015*). Therefore, our work draws attention to two distinct types of escape: flight to escape the environment as seen by jumps induced by pan-neuronal activation of dlPAG cells (*Figure 2I*) and flight to safer regions within the environment, as shown by l/vlPAG CCK+ cell activation (*Figures 2–3 and 6*). Further studies are needed to identify how l/vlPAG CCK+ cells affect downstream targets to produce escape as dissecting region-to-region connectivity can extend insight into defense circuits (*Bang et al., 2022*).

## Complementing columnar functional organization

Columnar organization in the PAG is supported by functional and anatomical similarities along the rostrocaudal axis. Broad activation of the dorsolateral column induces flight, hypertension, and tachycardia, while activation of the ventrolateral column induces freezing, hypotension, and bradycardia (*Keay and Bandler, 2015*). Neurotransmitter and receptor expression profiles and afferent/efferent connections are also generally (but not always) homogeneous within a single column along the anteroposterior axis (*Silva and McNaughton, 2019*).

However, to achieve a more complete understanding of PAG function, there may be other considerations in addition to columnar organization. There are exceptions to homogeneity along the anterior–posterior axis of columnar boundaries in PAG afferent and efferent connectivity. For example, adrenergic and noradrenergic medullary afferents preferentially target the rostral vlPAG and the central amygdala receives input from cells highly concentrated in the caudal but not rostral vlPAG (*Silva and McNaughton, 2019*). Furthermore, there are genetic markers that do not span an entire column; GABA-immunopositive cells are more prevalent in caudal than rostral cat vlPAG (*Barbaresi, 2005*). Tachykinin-1 (i.e., tac1), which is a marker of substance P-expressing cells, broadly spans multiple columns rostrally but concentrates in dorsolateral and ventrolateral columns caudally in the rat (*Liu and Swenberg, 1988*). Expression of rat endocannabinoid and glycine receptors also varies rostrocaudally, becoming more present in caudal PAG (*Araki et al., 1988*; *Herkenham et al., 1991*; *Silva and McNaughton, 2019*). It is likely that exploration of these genetically defined PAG populations will reveal novel insights. Indeed, inhibition of lPAG VGAT and lPAG VGlut2+ neurons impairs the chase and attack of prey, respectively (*Yu et al., 2021*), and glutamatergic vlPAG neurons project to the medulla to control freezing (*Tovote et al., 2016*). Moreover, l/vlPAG tac1+ cells have been shown to specifically control itching behavior (*Gao et al., 2019*), further supporting the value of investigating sparse genetically defined PAG populations.

Recent work has shown that examination of genetic diversity can unveil deep and novel understanding even in well-studied regions such as the amygdala. Accordingly, work from numerous groups dissecting glutamatergic basolateral amygdala cells based on genetic markers and projection targets has revealed the region's complex control of anxiety and valence processing (*Felix-Ortiz et al., 2013*; *Kim et al., 2016*; *Tye et al., 2011*).

In this study, using a genetic approach, we uncovered a sufficient and critical role of CCK+ cells in controlling threat avoidance. CCK-expressing cells are one among many largely uncharacterized genetically defined populations in the PAG (*Yin et al., 2014*), and here we outline a framework to assess how a single cell type may contribute to the vast constellation of behaviors controlled by the PAG. These results highlight that the molecular identity of PAG cells can lend key insight into functional motifs that govern how the PAG produces defensive responses and may serve as an additional axis of functional organization, complementing the well-established anatomical columnar PAG divisions.

# Materials and methods

## Materials availability

This study did not generate new unique reagents.

## Mice

*Cck-ires-cre* Mice (Jackson Laboratory Stock No. 012706) and wild-type C57BL/6J mice (Jackson Laboratory Stock No. 000664) were used for all experiments. Male and female mice between 2 and 6 months of age were used in all experiments. Mice were maintained on a 12 hr reverse light–dark cycle with food and water ad libitum. Sample sizes were chosen based on previous behavioral optogenetic studies on defensive behaviors, which typically use 6–15 mice per group. All mice were handled by experimenters for a minimum of 5 days prior to any behavioral task.

## Rats

Male Long–Evans rats (250–400 g) were obtained from Charles River Laboratories and were individually housed on a standard 12 hr reverse light–dark cycle with food and water ad libitum. Rats were only used as a predatory stimulus presentation. Rats were handled for several weeks prior to being used and were screened for low aggression to avoid attacks on mice. No attacks on mice were observed in this experiment.

## Method details

### Viral vectors

#### Optogenetics

The following AAV vectors were used in optogenetic experiments and were purchased from Addgene (Watertown, MA):

> AAV9-Ef1a-DIO EYFP (Addgene, 27056-AAV9).
> AAV9-EF1a-double floxed-hChR2(H134R)-EYFP-WPRE-HGHpA (Addgene, 20298-AAV9).
> AAV2.CMV.HI.eGFP-Cre.WPRE.SV40 (Addgene, 105545-AAV2).
> AAV9-hSyn-hChR2(H134R)-EYFP (Addgene, 26973-AAV9).
> AAV9-hSyn-EGFP (Addgene, 50465-AAV9).
> AAV9-FLEX-Arch-GFP (Addgene, 22222-AAV9).

#### Chemogenetics

The following AAVs, used in chemogenetic experiments, were purchased from Addgene:

> AAV8-hSyn-DIO-hM4D(Gi)-mCherry (Addgene, 44362-AAV8).
> AAV8-hSyn-DIO-mCherry (Addgene, 50459-AAV8).

#### Fiber photometry

The following AAVs, used in fiber photometry experiments, were purchased from Addgene:

> AAV9.Syn.Flex.GCaMP6f.WPRE.SV40 (Addgene, 100833-AAV9).
> AAV9.Syn.GCaMP6f.WPRE.SV40 (Addgene, 100837-AAV9).

## Surgeries

Ten-week-old mice were anesthetized with 1.5–3.0% isoflurane and affixed to a stereotaxic apparatus (Kopf Instruments). A scalpel was used to open an incision along the midline to expose the skull. After performing a craniotomy, 40 nL of virus was injected into the lateral and ventrolateral (l/vlPAG, unilateral and counterbalanced for optogenetic activation and fiber photometry experiments, bilateral for inhibition experiments) using a 10 μL Nanofil syringe (World Precision Instruments) at 40 nL/min. Affixed to the syringe is a 33-gauge beveled needle, and the bevel was placed to face medially. The syringe was slowly retracted 11 min after the start of the infusion. For l/vlPAG, infusion location measured as anterior–posterior, medial–lateral, and dorsoventral from bregma were –4.92 mm, ±1.25 mm, and –2.88 mm using a 15° angle. For dlPAG, –4.75 mm, –0.45 mm, and –1.9 mm using no angle. For chemogenetic experiments, mice received 40 nL of AAV8-hSyn-DIO-hM4D(Gi)-mCherry or

AAV8-hSyn-DIO-mCherry bilaterally. For optogenetic activation of CCK-l/vlPAG cells, 40 nL of AAV9-Ef1a-DIO-EYFP or AAV9-EF1a-DIO-ChR2(H134R)-EYFP-WPRE-HGHpA was delivered unilaterally to the l/vlPAG (counterbalancing left or right l/vlPAG) of CCK-cre mice. For optogenetic activation of synapsin-expressing l/vlPAG neurons, 40 nL of a viral cocktail (1:1) containing AAV2.CMV.HI.eGFP-Cre.WPRE.SV40 and AAV9-EF1a-DIO-hChR2(H134R)-EYFP-WPRE-HGHpA or a viral cocktail containing AAV2.CMV.HI.eGFP-Cre.WPRE.SV40 and AAV9-Ef1a-DIO EYFP was delivered to the l/vlPAG of wild-type mice. For optogenetic activation of synapsin-expressing dlPAG neurons, 40 nL of AAV9-hSyn-EGFP or AAV9-hSyn-ChR2(H134R)-EYFP was delivered unilaterally to the dlPAG of wildtype mice. For optogenetic inhibition of CCK-l/vlPAG cells, 40 nL of AAV9-Ef1a-DIO EYFP or AAV9-FLEX-Arch-GFP was delivered bilaterally to l/vlPAG. Mice used in optogenetic experiments received a fiber optic cannula (0.22 NA, 200 mm diameter; Doric Lenses) 0.2 mm above viral infusion sites. For photometry recordings of CCK-l/vlPAG cells, 40 nL of AAV9.Syn.Flex.GCaMP6f.WPRE.SV40 was injected into the l/vlPAG of CCK-cre mice and an optical fiber was implanted (0.48 NA, 400 mm diameter; Neurophotometrics) 0.2 mm above the injection site. For recordings of synapsin-expressing l/vlPAG cells, the same procedure was repeated using AAV9.Syn.GCaMP6f.WPRE.SV40 in wildtype mice. Dental cement (The Bosworth Company, Skokie, IL) was used to securely attach the fiber optic cannula to the skull. Half the mice in each cage were randomly assigned to YFP/mCherry control or ChR2/Arch/hM4di groups. Only mice with opsin expression restricted to the intended targets were used for behavioral assays.

## NeuN immunostaining

Fixed brains were kept in 30% sucrose at 4°C overnight, and then sectioned on a cryostat (40 μm slices). Sections were washed in PBS and incubated in a blocking solution (3% normal donkey serum and 0.3% Triton-X in PBS) for 1 hr at room temperature. Sections were then incubated at 4°C for 16 hr with polyclonal anti-NeuN antibody made in rabbit (1:500 dilution) (CAT# NBP1-77686SS, Novusbio) in blocking solution. Following primary antibody incubation, sections were washed in PBS three times for 10 min per wash, and then incubated with anti-rabbit IgG (H+L) antibody (1:1000 dilution) conjugated to Alexa Fluor 594 (red) (CAT# 8889S, cellsignal.com) for 2 hr at room temperature. Sections were washed in PBS three times for 10 min per wash, incubated with DAPI (1:50,000 dilution in PBS), washed again in PBS, and mounted in glass slides using PVA-DABCO (Sigma). Sections were imaged at ×20 magnification using a ZEISS LSM 900 confocal microscope. All imaging were done using standardized laser settings, which were held constant for samples from the same experimental dataset. For each animal, the dlPAG, lPAG, and vlPAG were imaged at two different sites within each region. CCK-GFP+ and NeuN+ cells were quantified for each site using ImageJ software (National Institutes of Health, Bethesda, MD). All CCK-GFP+ cells were also NeuN+. Within each region, quantified CCK-GFP+ and NeuN+ cells were totaled. To calculate the percentage of CCK-GFP+ neurons in the l/vlPAG, we divided CCK-GFP+ cells quantified in both lPAG and vlPAG by NeuN+ cells quantified in lPAG and vlPAG.

## vGlut2 immunostaining

Fixed brains were kept in 30% sucrose at 4°C overnight and then sectioned on a cryostat (40 μm slices). Sections were washed in PBS-T (0.3% Triton-X) and incubated in a blocking solution (5% normal donkey serum and 0.3% Triton-X in PBS) for 1 hr at room temperature. Sections were then incubated at 4°C for 16 hr with polyclonal anti-VGLUT2 antibody (#AGC-036, Alomone Labs) made in rabbit (1:500 dilution) in blocking solution. Following primary antibody incubation, sections were washed in PBS-T three times for 10 min per wash and then incubated with anti-rabbit IgG (H+L) antibody (1:1000 dilution) conjugated to Alexa Fluor 594 (red) (CAT# 8889S, cellsignal.com) in blocking solution for 2 hr at room temperature. Sections were washed in PBS-T three times for 10 min per wash, incubated with DAPI (1:50,000 dilution in PBS), and washed again in PBS-T and mounted in glass slides using PVA-DABCO (Sigma). Sections were imaged at ×10 magnification using a ZEISS LSM 900 confocal microscope. All imaging were done using standardized laser settings, which were held constant for samples from the same experimental dataset. For each animal, l/vlPAG was imaged and quantified using ImageJ software (National Institutes of Health) for DAPI+, vGlut2+, GFP+, and GFP+/vGlut2+ cells. Fraction of vGlut2-labeled cells also GFP-labeled was calculated as (vGlut2+ and GFP+)/vGlut2+. Fraction of GFP-labeled cells also vGlut2-labeled was calculated as (vGlut2+ and GFP+)/GFP+.

## In situ hybridization

Mice were euthanized with 5% isoflurane followed by cervical dislocation. Brains were harvested and snap-frozen in 2-methylbutane at −20°C and tissue was sectioned at 18 μm. The workflow was performed in accordance with the manufacturer's protocol for the RNAScope Multiplex Fluorescent Assay (Advanced Cell Diagnostics, Newark, CA). Riboprobes selective for sequences were labeled as follows: CCK (Mm-CCK, Cat# 402271) or VGLUT2 (Mm-Slc17a6-C3, Cat# 319171-C3). Images were obtained with a ZEISS LSM 900 confocal microscope at ×40.

## Behavior video capture

All behavior videos were captured at 30 frames/s in standard definition (640 × 480) using a Logitech HD C310 webcam. To capture fiber-photometry synchronized videos, both the calcium signal and behavior were recorded by the same computer using custom MATLAB scripts that also collected timestamp values for each calcium sample/behavioral frame. These timestamps were used to precisely align neural activity and behavior.

## Behavioral timeline

The order of behavior assays for optogenetic and fiber photometry experiments is as follows (if applicable): open field, EPM, LTE, real-time place preference, toy rat exposure, real rat exposure, and pupil dilation. The order of behavior assays for chemogenetic experiments is as follows: toy rat exposure, live rat exposure, hot plate, and cued fear conditioning.

## Light delivery for optogenetics

For all ChR2 experiments, blue light was generated by a 473 nm laser (Dragon Lasers, Changchun Jilin, China) at 3–3.5 mW with the exception of pupil dilation recordings in which a 1.5 mW power was used to avoid overt escape responses during head fixation. For the Arch experiment, green light was generated by a 532 nm laser (Dragon Lasers) and bilaterally delivered to mice at 5–6.6 mW. A Master-8 pulse generator (A.M.P.I., Jerusalem, Israel) was used to drive the blue laser at 20 Hz. This stimulation pattern was used for all ChR2 experiments. The laser output was delivered to the animal via an optical fiber (200 mm core, 0.22 numerical aperture, Doric Lenses, Canada) coupled to the fiberoptic implanted on the animals through a zirconia sleeve.

## Open-field assay with optogenetics

The open field is a square arena (34 × 34 × 34 cm) illuminated to 105 lux. Mice had no prior experience in the arena prior to exposure. Exposures are 8 min total, with alternating 2 min epochs of laser off or on (OFF, ON, OFF, ON).

## LTE assay with optogenetics

The LTE assay was carried out across two consecutive days in a square (47 × 47 × 36 cm) arena. The arena is illuminated to 80 lux and contains a dark burrow (7 × 13 × 11 cm, 2 lux) in one corner. On day 1, mice are habituated to the entire arena for 10 min. Only mice that spent more time in the burrow compared to the other three corners continued to day 2. Out of 69 mice from all cohorts, only 6 were excluded due to not showing preference for the burrow. On day 2, a transparent barrier is placed in the corner opposite of the burrow to create a holding zone. Mice were placed in the arena for 1 min as a context reminder prior to placement in the holding zone. Then, 10 trials were carried out, with 5 laser-off and five laser-on trials interleaved. Prior to all trials, mice were confined to the holding zone for 15 s prior to barrier removal. For light-on trials, laser is delivered for the latter five of the 15 s and continues until the end of the trial. The start of the trial is marked by barrier removal and ends when the mouse enters the burrow or when 60 s have passed. If mice enter the burrow, they are given 10 s in the burrow prior to being returned to the holding zone. If mice do not enter the burrow, they are immediately returned to the holding zone. The procedure is similar between ChR2 activation and Arch inhibition in the LTE, except for laser wavelength and pulse length. Mice were handled for a minimum of days. This habituation to handling served to decrease any potential anxiety or stress caused by the handling involved in placing the mouse in the holding zone.

## Place aversion test with optogenetics

Mice were placed in a two-chamber context (20 × 42 × 27 cm) for 10 min to freely explore the environment. Both chambers are identical. The following day, mice were introduced into the two-chamber context and blue laser was delivered to the l/vlPAG of CCK-cre mice expressing either ChR2 or YFP (20 Hz, 5 ms pulses) when they occupied one of the chambers. Laser stimulation was only delivered during exploration of the stimulation chamber. The amount of time mice explored both chambers was tracked across both the baseline and stimulation epochs.

## EPM with optogenetics and fiber photometry

The arms of the EPM were 30 × 7 cm. The height of the closed arm walls was 20 cm. The maze was elevated 65 cm from the floor and was placed in the center of the behavior room away from other stimuli. Arms were illuminated to 8–12 lux. Mice were placed in the center of EPM facing a closed arm. For optogenetic experiments, blue light was delivered in alternating 2 min epochs for five epochs, totaling 10 min exposure. The fifth epoch was excluded from analyses. ChR2 activation was only run in l/vlPAG-CCK group as overt freezing in the l/vlPAG-syn group would not be informative and the assay is not feasible with robust jumping exhibited by ChR2 activation of dlPAG-syn mice. For fiber photometry recordings, mice were free to explore the EPM for 10 min.

## Pupil size measurements with optogenetics

Pupil size was measured with the same set-up and methods described previously (*Lovett-Barron et al., 2017*) Briefly, a camera (AVT Manta, G-032B) coupled to a 24 mm/F1.4 lens was used to image the eye under infrared illumination (Thorlabs M780F2). Video was acquired at 60 Hz using pymba, a Python wrapper for AVT camera control. Frame acquisition times and the behavioral task were synchronized with a National Instruments DAQ (NIPCIe-6323). Pupil size was measured from the video using custom-written MATLAB scripts. Each trial lasted 30 s. A 473 nm laser (1.5 mW) was delivered to the l/vlPAG of CCK-cre mice at 20 Hz, 5 ms pulses for 10 s following a 10 s baseline recording. Another 10 s were recorded post-stimulation.

## Live rat exposure assay with optogenetics, chemogenetics, and fiber photometry

We used a long rectangular chamber (70 × 25 × 30 cm). Mice were acclimated to this environment for at least 2 days for 10 min each day. During rat exposure, a live rat is restrained to one end of the chamber using a harness attached to a cable with one end taped to the chamber wall. As a behavioral control, we exposed mice to a toy rat (similar in shape and size to a live rat) to assess behavior elicited by visually similar stimuli without actual predatory threat. For the optogenetic experiment, mice were presented with the toy rat for one trial prior to exposure to live rat one day after. On each day, 473 nm laser alternated off or on in 2 min epochs for five epochs, totaling in a 10 min trial. Only the first four epochs (OFF, ON, OFF, ON) were included in analyses. For the chemogenetic experiment, mice were exposed to two toy rat trials on consecutive days followed by two live rat exposures on consecutive days. Mice either received CNO or saline prior to exposures. All trials were 10 min. For fiber photometry recordings, all mice underwent toy rat exposure followed by live rat exposure the following day for 10 min each.

## Chemogenetics

Mice used for all chemogenetic experiments (with the exception of cued fear conditioning) were exposed to each threat and control stimuli twice, once following treatment with saline and once following treatment with CNO (10 mg/kg, injected intraperitoneally) 40 min prior to the experiment. Only one control or threat-exposure assay was performed per day with each mouse. For cued fear conditioning, all mice received CNO prior to training.

## Heated plate assay with chemogenetics

Heated plate assay was performed on top of a metallic heating plate (14 × 14 cm) (Faithful Magnetic Stirrer model SH-3) across two sequential days on mice expressing AAV8-hSyn-DIO-mCherry or AAV8-hSyn-DIO-hM4D(Gi)-mCherry. On both days, mice received either CNO (i.p., 10 mg/kg) or saline 40 min prior to heated plate exposure. The order of drugs was counterbalanced. Plate was heated to

50°C and enclosed by tall, transparent walls (14 × 14 × 24 cm). Mice were closely monitored for pain response. The latency to display a pain-related reaction (hind paw lick or jump) was recorded. All mice showed pain responses within 30 s. After pain response, mice were immediately removed from assay. Pain latency was measured as CNO minus SAL (CNO – SAL).

## Fear conditioning with chemogenetics

Standard mouse fear conditioning chambers from Coulbourn Instruments were used. The chamber dimensions were 7 × 7 × 12 inches (width × length × height). Cued fear conditioning was performed across two sequential days on mice expressing AAV8-hSyn-DIO-mCherry or AAV8-hSyn-DIO-hM4D(Gi)-mCherry. Mice were habituated to the fear conditioning room for 2 days prior to the start of the experiment. Mice were handled by the same experimenter during habituation as well as days 1–2 of the experiment. On day 1 (training), all mice received CNO (i.p., 10 mg/kg). Then, 40 min later, mice were exposed to a context consisting of metal bar flooring and bare gray walls, and cleaned with 70% ethanol. Training context was illuminated with warm-colored white lighting at 40 lux. Mice received 10 CS-US pairings pseudo-randomly presented across a 14 min trial. CS was a 10 s pure tone at 70 dB and US was a 1 s, 0.6 mA shock. CS and US coterminated. The first CS-US pairing began 100 s after the start of the trial. Following training, mice were returned to their home cage. On day 2 (retrieval test), mice were exposed to a context consisting of rounded white walls and gray smooth flooring and cleaned with Strike-Bac (Chino, CA). These changes assure that visual and olfactory cues are different between the training and testing contexts. Retrieval test context was illuminated with blue lighting at 40 lux. Mice received 10 CS-only presentations (same CS configuration as training) across a 14 min trial. Freezing during CS presentation was scored in an automated manner using FreezeFrame 5 (Actimetrics, IL). Freeze bouts were at minimum 0.33 s in duration. This same 0.33 s freeze bout duration was used throughout all assays. Percent time freezing showed a correlation of over 0.95 when calculated using either 0.33 or 1 s minimum bout duration.

## Fiber photometry

Photometry was performed as described in detail previously (*Kim et al., 2016*). Briefly, we used a 405 nm LED and a 470 nm LED (Thorlabs, M405F1 and M470F1) for the $Ca^{2+}$-dependent and $Ca^{2+}$-independent isosbestic control measurements. The two LEDs were band-pass filtered (Thorlabs, FB410-10 and FB470-10) and then combined with a 425 nm long-pass dichroic mirror (Thorlabs, DMLP425R) and coupled into the microscope using a 495 nm long-pass dichroic mirror (Semrock, FF495-Di02-25 3 36). Mice were connected with a branched patch cord (400 mm, Doric Lenses, QC, Canada) using a zirconia sleeve to the optical system. The signal was captured at 20 Hz (alternating 405 nm LED and 470 nm LED). To correct for signal artifacts of a nonbiological origin (i.e., photo-bleaching and movement artifacts), custom MATLAB scripts leveraged the reference signal (405 nm), unaffected by calcium saturation, to isolate and remove these effects from the calcium signal (470 nm).

## Perfusion and histological verification

Mice were anesthetized with Fatal-Plus (i.p., Vortech Pharmaceuticals, Dearborn, MI) and transcardially perfused with PBS followed by 4% paraformaldehyde. Extracted brains were stored for 12 hr at 4°C in 4% paraformaldehyde before transfer to 30% sucrose for a minimum of 24 hr. Brains were sectioned into 40 μm coronal slices in a cryostat, washed in PBS, and mounted on glass slides using PVA-DABCO. Images were acquired using a Keyence BZ-X microscope (Keyence Corporation of America, Itasca, IL) with a ×4, ×10, or ×20 air objective.

## Behavioral quantification

To extract the pose of freely behaving mice in the described assays, we implemented DeepLabCut (*Nath et al., 2019*), an open-source convolutional neural network-based toolbox, to identify mouse nose, ear, and tail base xy-coordinates in each recorded video frame. These coordinates were then used to calculate velocity and position at each time point, as well as classify behaviors such as threat approaches, escape runs, stretch-attend postures, and freeze bouts in an automated manner using custom MATLAB scripts. Freezing was defined as epochs when head and tailbase velocities fell below 0.25 cm/s for a period of 0.33 s. 'Stretch-attend postures' were defined as epochs for which (1) the distance between mouse nose and tail base exceeded a distance of approximately 1.2 mouse body

lengths and (2) mouse tail base speed fell below 1 cm/s. Approach and escape were defined as epochs when the mouse moved, respectively, toward or away from the rat at a velocity exceeding a minimum threshold of 3 cm/s. All behaviors were manually checked by the experimenters for error.

## Statistics

Unpaired $t$-tests of ON minus OFF or CNO minus SAL transformations were used unless otherwise stated. Normality of data was tested with the Lilliefors test. Two-tailed $t$-tests were used throughout with $\alpha = 0.05$. Correlations were calculated using Pearson's method. Asterisks in the figures indicate the p-values for the post-hoc test. Standard error of the mean was plotted in each figure as an estimate of variation. Multiple comparisons were adjusted with the false discovery rate method.

## Behavioral cohort information

Each primary experiment included at least one replication cohort. Each mouse was only exposed to each assay once as defensive behavior assays cannot be repeated. Thus, there are no technical replicates. No outliers were found or excluded. All mice were used. Sample sizes were determined based on comparisons to similar published papers.

For chemogenetic and optogenetic experiments, mice in each cage were randomly allocated to control (mCherry or YFP-expressing mice) or experimental conditions (hM4Di, ChR2, or Arch-expressing mice). Data collection was done blinded to treatment group in mice. For mouse fiber photometry neural activity recordings, all data were obtained from subjects in identical conditions, and thus they were all allocated to the same experimental group. There were no experimentally controlled differences across these subjects and, thus, there were no 'treatment groups'.

## Data and code availability

Custom analysis scripts are available at https://github.com/schuettepeter/l-vlPAG_ActiveAvoidance; (*La-Vu, 2022* copy archived at swh:1:rev:e77591414eade868e6f5459df18b6ce777d3905a). Data is available at https://doi.org/10.5068/D12Q32.

## Acknowledgements

We were supported by the NIMH (R00 MH106649 and R01 MH119089 to AA and F31 MH121050-01A1 to ML-V); the Achievement Rewards for College Scientists Foundation, Los Angeles Chapter (to ML-V); the Brain and Behavior Research Foundation (22663, 27654, and 27780 to AA, FMCVR, and WW respectively); the NSF (NSF-GRFP DGE-1650604 to PS); the UCLA Affiliates fellowship (to PS); the Hellman Foundation (to AA); FMCVR was supported by FAPESP grants 2015/23092-3 and 2017/08668-1. We thank Profs. S Correa, KM Wassum, and M Fanselow for helpful discussions. We thank HT Blair and KM Wassum for providing rats.

## Additional information

### Funding

| Funder | Grant reference number | Author |
|---|---|---|
| National Institute of Mental Health | R00 MH106649 | Avishek Adhikari |
| National Institute of Mental Health | R01 MH119089 | Avishek Adhikari |
| National Institute of Mental Health | F31 MH121050-01A1 | Mimi Q La-Vu |
| Achievement Rewards for College Scientists Foundation | | Mimi Q La-Vu |
| Brain and Behavior Research Foundation | 22663 | Avishek Adhikari |

| Funder | Grant reference number | Author |
|---|---|---|
| Brain and Behavior Research Foundation | 27654 | Fernando MCV Reis |
| Brain and Behavior Research Foundation | 27780 | Weisheng Wang |
| UCLA Health System | UCLA Affiliates fellowship | Peter J Schuette |
| Hellman Foundation | | Avishek Adhikari |
| Fundação de Amparo à Pesquisa do Estado de São Paulo | 2015/23092-3 | Fernando MCV Reis |
| Fundação de Amparo à Pesquisa do Estado de São Paulo | 2017/08668-1 | Fernando MCV Reis |
| National Science Foundation | NSF-GRFP DGE-1650604 | Peter J Schuette |

The funders had no role in study design, data collection and interpretation, or the decision to submit the work for publication.

### Author contributions

Mimi Q La-Vu, Conceptualization, Formal analysis, Funding acquisition, Methodology, Project administration, Supervision, Writing – original draft, Writing – review and editing; Ekayana Sethi, Conceptualization, Investigation, Methodology; Sandra Maesta-Pereira, Formal analysis, Investigation, Software; Peter J Schuette, Formal analysis, Software; Brooke C Tobias, Weisheng Wang, Anita Torossian, Investigation, Methodology; Fernando MCV Reis, Conceptualization, Investigation; Amy Bishop, Catherine M Cahill, Investigation; Saskia J Leonard, Lilly Lin, Methodology; Avishek Adhikari, Conceptualization, Funding acquisition, Project administration, Supervision, Writing – review and editing

### Author ORCIDs

Brooke C Tobias http://orcid.org/0000-0003-2043-9523
Avishek Adhikari http://orcid.org/0000-0002-9187-9211

### Ethics

All procedures conformed to guidelines established by the National Institutes of Health and have been approved by the University of California, Los Angeles Institutional Animal Care and Use Committee (protocol #2017-011) .

### Decision letter and Author response

Decision letter https://doi.org/10.7554/eLife.77115.sa1
Author response https://doi.org/10.7554/eLife.77115.sa2

## Additional files

### Supplementary files

• Transparent reporting form

### Data availability

Data is available on Dryad: https://doi.org/10.5068/D12H4C.

The following dataset was generated:

| Author(s) | Year | Dataset title | Dataset URL | Database and Identifier |
|---|---|---|---|---|
| La-vu M | 2022 | Sparse genetically-defined neurons refine the canonical role of periaqueductal gray columnar organization | https://doi.org/10.5068/D12H4C | Dryad Digital Repository, 10.5068/D12H4C |

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
