## [Editor Report]

The article is a tour de force examination of the role of PAG CCK neurons in threat. It is exemplary in the use of a variety of high- and low-threat tasks as well as gain and loss of CCK function approaches, and reporting of distinct behaviors. The results reported will be of significant benefit for those studying the behavioral and neural mechanisms of learned and unlearned fear and threat, and decision-making in threatening situations.

---

## [Decision Letter]

**Decision letter after peer review:**

Thank you for submitting your article "A genetically-defined population in the lateral and ventrolateral periaqueductal gray selectively promotes flight to safety" for consideration by *eLife*. Your article has been reviewed by 3 peer reviewers, including Mihaela D Iordanova as Reviewing Editor and Reviewer #1, and the evaluation has been overseen by Laura Colgin as the Senior Editor.

Essential revisions:

1) The optogenetic and photometry data need to be better reconciled, including accounting for why the activity is high in safe zones away from threat but stimulation leads to aversion. Given the aversive effect of cck neuron stimulation in the RTPP that is reported in the manuscript, one would hypothesize that stimulating those neurons in any other context will also evoke aversiveness. So this begs the question: why under unperturbed conditions as shown by photometry, neuronal activity of CCK cells is enhanced in low threat areas, while the same cells mediate aversion when optogenetically activated? Essentially, this would render the safe zone aversive, which appears counterintuitive.

2) More in-depth discussion of past literature: Prior data in the field provide strong support that dPAG, not vPAG, mediates flight, as well as a role for vlPAG in prediction error in fear.

3) Current statistical reporting is minimal. The authors need to be back each result statement by statistics in the text. Asterisks and p values denoting significance are not sufficient.

4) The conclusion that CCK neuronal activity does not affect learning is somewhat premature given the data. The result could be explained in terms of ceiling levels of responding, so this conclusion should be softened.

5) Methodological concerns:

a. A discussion of the effect of sub-selecting high anxiety mice that preferred the burrow during habituation as well as creating a high-anxiety situation by moving the animals every 60s in the LTE task on CKK neuron activity.

b. Include an assessment of which part of the apparent CCK clusters was targeted. This should include at the very least fiber positions, and ideally also viral spread maps.

c. Include information on cell counting procedure, data reduction and analysis.

d. Add information on how many VGluT2^+^ neurons were also CCK^+^.

e. Add information on parametric/non-parametric nature of the data, and usage of the adequate stats.

f. Information of conditioning setup and experimental design is missing.

g. Explain the choice of a very short and unconventional freezing duration criterion, as well as varying freezing criteria across behavioral tests.

h. Figure 8 J and K are missing the number of mice in the legend, and single data points on the graphs.

i. Revise the pupil dilation experiment either interpretation-wise or add new data. Our concern is, that if radically different stimulation parameters are used, such as intentionally by the authors here to get rid of behavioural responses, how can we assume that the results can be generalized over apparently different defensive states? A comparison of the behavioural responses for the two intensities (high intensity, as reported for behaviour, and low intensity, as for the pupil dilation) in the same animals can help clarify this issue.

j. Stimulation in the EPM follows a 2min ON 2min OFF order. It would be important to know when stimulation occurs – when the animal is in the open or closed arm and whether this correlates with the percent time spent in the open arm. To explain, if stim of CCK neurons is aversive as suggested by the RPTP task then there could be a summation of aversiveness when stimulation occurs in the open vs closed arm.

---

## [Author Response]

Essential revisions:1) The optogenetic and photometry data need to be better reconciled, including accounting for why the activity is high in safe zones away from threat but stimulation leads to aversion. Given the aversive effect of cck neuron stimulation in the RTPP that is reported in the manuscript, one would hypothesize that stimulating those neurons in any other context will also evoke aversiveness. So this begs the question: why under unperturbed conditions as shown by photometry, neuronal activity of CCK cells is enhanced in low threat areas, while the same cells mediate aversion when optogenetically activated? Essentially, this would render the safe zone aversive, which appears counterintuitive.

We thank the Reviewer for this important point, and appreciate the opportunity to more wholly reconcile our optogenetic and photometry findings. We agree with the Reviewers that at first glance the optogenetic and photometry data may appear to be conflicting. However, both datasets can be interpreted as a reflection that CCK cells are controlling avoidance states during threat exposure. During exploration of the rat assay, mice alternate between threat approach and threat avoidance behavioral states. In the approach state, mice decrease their distance to the rat, by approaching it. During the avoidance state, mice increase distance to the threat and engage in evasive actions, such as escape. The photometry data show CCK cells are active during both escape and during moments of increased distance to threat, and thus seem to reflect a threat avoidance state. In line with this interpretation, CCK activation increases distance to threat. According to this interpretation, higher CCK activity creates higher distance to threat, and thus it would be expected to elicit escape and to be higher during avoidance of the rat. L/vlPAG CCK cell activity also decreases during approach to the rat because during these moments the mouse is in the threat approach state, not in the threat avoidance state.

We also note that the ‘safe zone’ in the rat assay (see Figure 7c) is not comfortable, but rather it is safe relative to the more dangerous ‘threat zone’ near the rat. We thus argue that this ‘safe zone’ away from the rat is indeed still aversive as noted by the Reviewer, (though less aversive than the threat zone). Thus, if possible, the mouse would likely escape from this less aversive ‘safe zone’ and go to a truly danger-free location, but there is no better location for the mouse to choose, so it continues to stay as far as possible from the rat within the confines of the box. Higher CCK activity thus does not elicit escape from the current location, but rather signals the motivation to escape to the safest place available. If the current location is aversive but is still the safest possible place then higher CCK activity would not change the mouse’s location, even though escape to a truly safe place would be desirable despite being impossible.

We interpret the optogenetically-induced aversion and flight to safety as well as the enhanced neuronal activity in CCK cells in low threat areas to be, respectively, driving and reflecting avoidance. As we show that activation of CCK cells produced strong thigmotaxis in the open field, urgent flight to a dark burrow, and bias toward the closed arms in the epm, we interpret that CCK activation under low-threat conditions induces evasion of bright, open spaces. Relatedly, CCK activation in high-threat conditions (predator exposure) caused evasion of the most salient perceived threat, the predator. Additionally, CCK inhibition increased time near the predator, further suggesting that CCK activity drives the threat avoidance state.

The Reviewer has highlighted that this interpretation was not previously made clear, and we have thus adjusted the wording of our interpretation throughout the manuscript as well as significantly expanded our interpretation in the Discussion section.

2) More in-depth discussion of past literature: Prior data in the field provide strong support that dPAG, not vPAG, mediates flight, as well as a role for vlPAG in prediction error in fear.

We agree with the Reviewer that these points are important and warrant further discussion. Accordingly, we have revised our Discussion to emphasize these points.

Regarding the role of dPAG in mediating flight, we emphasize that escape has historically been attributed to the dorsal PAG. We now mention that such robust and urgent flight has been reported in the dPAG by a number of studies including Jenck et al., (1995), Valenstein (1965), Perusini and Fanselow (2015), Del-Ben and Graeff (2009), Depaulis et al., (1992), and Fanselow (1991), and we conclude that our present findings suggest that the PAG role in escape extends beyond the dPAG column to the l/vlPAG.

Regarding the vlPAG role in prediction error, we discuss conceptual models described in McNally et al., (2011) and Herry and Johansen (2014) that suggested the PAG sends information regarding unconditioned stimulus information to the amygdala, and that this feature is critical for prediction error coding. Though Ozawa et al., (2017) and Walker et al., (2020) show that vlPAG neurons are indeed critical for prediction error coding, our CCK^+^ cell inhibition during conditioned fear acquisition did not alter freezing during acquisition nor retrieval. Therefore, we conclude that prediction error coding is carried out by other vlPAG cells in the conditioning protocol we used. However, as discussed in Review question 4 below, it is entirely possible that CCK cells can affect fear conditioning in other protocols that we did not test.

3) Current statistical reporting is minimal. The authors need to be back each result statement by statistics in the text. Asterisks and p values denoting significance are not sufficient.

We apologize for this oversight, and we have now included complete descriptive statistics throughout the main text and figure legends including name of statistical test, n for each group and exact p value.

4) The conclusion that CCK neuronal activity does not affect learning is somewhat premature given the data. The result could be explained in terms of ceiling levels of responding, so this conclusion should be softened.

We agree with this comment, and have now changed the text to include this consideration. However, please note that during retrieval freezing levels were around 80% (Figure 7, figure supplement 2), and thus freezing was not at ceiling levels of 100%. Nevertheless, we agree with the Reviewer that it certainly is possible that the CCK cells may have a role in fear learning when using different experimental parameters. We have changed the text to discuss this important possibility.

5) Methodological concerns:a. A discussion of the effect of sub-selecting high anxiety mice that preferred the burrow during habituation as well as creating a high-anxiety situation by moving the animals every 60s in the LTE task on CKK neuron activity.

We note that only 6 out of a total of 69 mice run in this assay were excluded for not showing a preference for the burrow during habituation. The vast majority of mice (91.3%) that underwent LTE habituation continued onto testing, thus the results are not derived from a subset of highly anxious mice. We removed these 6 mice, because if the mouse does not recognize the burrow as a safest region within the assay (and thus spends more time in the burrow than the other corners) prior to optogenetic stimulation, then the assay cannot be interpreted as a measure of flight to safety as the main metric is latency to enter the burrow.

It is true that the experimenter had to manually replace animals in the holding zone; however, mice were well-habituated to handling prior to the assay for a minimum of 5 days in order to decrease anxiety responses caused by handling by the experimenter. Furthermore, identical handling happened in both light OFF and ON trials, thus anxiety or changes in CCK cell activity caused by handling does not account for our bidirectional optogenetic effects.

We have added discussions in the text to include these important considerations.

b. Include an assessment of which part of the apparent CCK clusters was targeted. This should include at the very least fiber positions, and ideally also viral spread maps.

We targeted the entire CCK lPAG/vlPAG population, and thus did not investigate any subset of this cluster. These cells span a small volume of tissue, thus our experiments were able to consistently target the whole region containing lPAG/vlPAG CCK cells. We have added viral spread maps and fiber tip positions to more clearly demonstrate this point, as requested by the Reviewer.

c. Include information on cell counting procedure, data reduction and analysis.

We have provided information on post-histological procedures including confocal imaging parameters, cell counting methods within specific brain regions and how each metric was defined and calculated in the two Methods sections, “NeuN immunostaining” and “vGlut2 immunostaining.” For NeuN cell counting procedures, we imaged the dlPAG, lPAG, and vlPAG at two different sites within each region and quantified NeuN+ and CCK-GFP+ cell for each site. To calculate the percentage of neurons that were CCK-GFP+ in the l/vlPAG and dlPAG, we normalized CCK-GFP+ cells by NeuN+ cells. For vGlut2 cell counting procedure, we obtained confocal imaging of the l/vlPAG and quantified DAPI+ cells, vGluT2^+^ only cells, CCK-GFP+ only cells, and vGluT2^+^/GFP+ cells. To obtain the fraction of vGluT2^+^ cells that were also GFP+, we normalized vGluT2^+^/GFP+ by vGluT2^+^ only cells. To calculate the fraction of GFP+ cells that were also vGluT2^+^, we normalized vGluT2^+^/GFP+ by GFP+ only cells. All counting was done using the freely available ImageJ software.

d. Add information on how many VGluT2^+^ neurons were also CCK^+^.

We apologize for this oversight and appreciate the opportunity to provide more information. To address this comment we injected a cre-dependent virus encoding YFP in the l/vlPAG of CCK-cre mice, resulting in YFP expression in CCK^+^ cells. We then performed an antibody stain against the glutamatergic marker vGlut2. We performed the requested quantification and found that 9.6% of vGluT2^+^ cells in the l/vlPAG are also CCK-GFP+ (see Figure 1e). Additionally, we performed in situ hybridizations to confirm this result and found that 8.6% of vGlut2-expressing cells are also CCK-positive (Figure 1, figure supplement 1). Thus, two different methods of quantification provide converging evidence that CCK cells are a sparse population comprising less than 10% of l/vlPAG vGlut2 cells.

e. Add information on parametric/non-parametric nature of the data, and usage of the adequate stats.

We verified that the data was normal using the lilliefors test, and thus used parametric tests.

f. Information of conditioning setup and experimental design is missing.

We apologize for the lack of clarity and have provided more information on our fear conditioning procedures under the section ‘Fear conditioning with chemogenetics.’ We have also supplemented the caption for Figure 7, figure supplement 2 with additional information regarding specific parameters used during conditioning.

g. Explain the choice of a very short and unconventional freezing duration criterion, as well as varying freezing criteria across behavioral tests.

We thank the Reviewer for this astute observation and apologize for the oversight. While the minimum freezing bout duration scored using custom MATLAB scripts was 0.33 s, the minimum freezing bout scored using the FreezeFrame 5 software program for the fear conditioning experiment was 0.5 s. This oversight has been corrected and, now, freezing reported in all behavioral experiments has been scored using the 0.33 s minimum bout duration. Importantly, this rescoring did not alter the findings previously reported in the fear conditioning experiment (Figure 7, figure supplement 2).

We agree with the Reviewer that 0.33 s is a relatively short freezing bout duration compared to conventional reporting methods. Historically, minimum freezing bout lengths were greater because behavioral data was largely humanly scored, which generally limited minimum bout durations to 1 second.

However, for this manuscript, we aimed to leverage the sensitivity of automated scoring methods. It remains important, however, to note that our behavioral data does not vary appreciably due to differences in scoring methodology. Thus, we show that freezing duration (Author response image 1) and number of freeze bouts (Author response image 1) scored when minimum freeze bouts were set at 1 s or 0.33 s is highly correlated. Furthermore, we show that the median bout duration is 1.4 s and the average bout duration is approximately 2 s (Author response image 1; 682 freeze bouts).

**Author response image 1. sa2fig1:** Scatterplot shows the correlation between scored freeze duration (A) and scored freeze bouts (B) when minimum freezing bout duration is 1s and 0. 33 s. This behavioral data is derived from YFP mice undergoing predator exposure without laser manipulation (n=10). (C) Histogram shows the median freeze bout duration is 1.4 s and average bout duration is 2 s (n=682 freezes).

h. Figure 8 J and K are missing the number of mice in the legend, and single data points on the graphs.

We apologize for this oversight. The purpose of those plots was to show that CCK activity was higher far from the threat during either escape or approach to the rat. To show this effect, we had compared neural activity in the left and right side of the enclosure during those behaviors. While this approach produced a significant effect, during the revision we realized that this method arbitrarily separated the environment into two locations. Thus, now, to show the relationship between CCK activity and distance to the rat we show that CCK activity is negatively correlated with proximity to the threatening ra (Figure 8I)t. This method is better than the previous version of the plots because the correlation does not require separating arbitrarily the environment into two subregions, and is a more unbiased way of illustrating that CCK activity is lower when the mouse is closer to the rat, in both approach and escape. The n’s are n=9 syn mice and 13 CCK mice, and this information has now been added to the legend. We refrained from plotting individual data points for each of the spatial bins because this would result in a cluttered graph with an unreasonably large number of points in each plot (90 syn points and 130 CCK points, as there are 10 spatial bins in the correlation). Nevertheless, we are glad to add these extra data points if the reviewers feel they are necessary.

i. Revise the pupil dilation experiment either interpretation-wise or add new data. Our concern is, that if radically different stimulation parameters are used, such as intentionally by the authors here to get rid of behavioural responses, how can we assume that the results can be generalized over apparently different defensive states? A comparison of the behavioural responses for the two intensities (high intensity, as reported for behaviour, and low intensity, as for the pupil dilation) in the same animals can help clarify this issue.

Thank you for this comment and for the opportunity to clarify why we chose to employ a lower-intensity stimulation protocol for the optogenetic pupil dilation experiment. We first note that behaviors and pupil dilation were done, respectively, at 3.5 mW and 1.5 mW. These values are different, but we respectfully disagree with the Reviewer’s statement that these are “radically different stimulation parameters.” We also note that the same stimulation frequency and pulse length were used.

We used lower-intensity stimulation for the pupil diameter assay because 1.5 mW power did not elicit the overt escape responses observed at higher-intensity stimulation (3-3.5 mW). It is not possible to use high intensity stimulation in the head-fixed setup, as our preliminary attempts to do so showed that optogenetic stimulation of CCK cells at 3.5 mW induced vigorous escape motion, almost causing the headcap implant to be pulled out. Furthermore, lower-intensity stimulation allowed us to avoid movement-related arousal or stress (from attempting to escape but not being able to) that could have indirectly increased pupil dilation as a secondary effect and potentially confounded the measurements. Thus, our experiment shows that changes in pupil size are a consequence of the optogenetic stimulation, rather than a consequence of increased locomotion caused by the stimulation.

Although it is not possible to measure pupil size at 3.5 mW for the reasons outlined above, it is very probable that the pupil size would also increase at 3.5 mW. It is fairly unlikely that the pupil size would not be increased when the animal receives higher stimulation causing behavioral manifestations of escape. In our initial attempts we observed that clear behavioral escape responses were not elicited at 1.5 mW, which prompted us to increase the stimulation power to the minimum amount that induced observable behavioral changes.

Thus, indeed, as stated by the Reviewer, stimulation at different intensities may create distinct behavioral states. We have thus changed the interpretation of the data as requested, and now state that lower simulation is sufficient to induce physiological changes, while higher stimulation produces behavioral escape responses. This interpretation fits the data, and indicates small increases in l/vlPAG CCK cells induce the physiological pupil size change first, indicating higher alertness, which may serve as a useful pre-emptive preparation for eventual escape if CCK activity increases further.

j. Stimulation in the EPM follows a 2min ON 2min OFF order. It would be important to know when stimulation occurs – when the animal is in the open or closed arm and whether this correlates with the percent time spent in the open arm. To explain, if stim of CCK neurons is aversive as suggested by the RPTP task then there could be a summation of aversiveness when stimulation occurs in the open vs closed arm.

We thank the Reviewer for this important observation. It is true that with a fixed duration ON/OFF laser delivery protocol as was used here to apply optogenetic manipulation to the elevated plus maze assay, we are unable to control whether laser onset coincides with occupation of the open or closed arms. As we show that stimulation of CCK cells is aversive in the real-time place test assay, it is possible that stimulation of CCK cells while in a certain arm may lead to a summation of aversion of said arm. To address this, we show in Author response image 2 that a majority (60 ± 10%) of laser onsets occur while ChR2 mice (n=10) occupy a closed arm, and the same mice also spend a majority of laser delivery in the closed arms (68.16 ± 3.39%). Thus, while a majority of laser onsets occurred while mice occupied the closed arms, mice also spent a majority of time in the closed arms, suggesting that arm occupancy during laser onset did not result in a summation of aversion to the closed arms.

**Author response image 2. sa2fig2:** Data from ChR2 mice (n=10) receiving stimulation to l/vlPAG-CCK cells during elevated plus maze assay. (A) Bar plot showing mice occupied a closed arm during 60% of laser onsets. Each 8-min EPM trial consisted of four 2-min epochs with alternating laser delivery (OFF-ON-OFF-ON) resulting in two laser onsets per trial. (B) Mice spent 68.16% of time in the closed arms during laser delivery; thus mice spent the majority of time in the closed arms, indicating that arm occupancy during laser onset did not result in aversion of said arm. Mean ± SEM.